

# User-oriented hydrological indices for early warning system. Validation using post-event surveys: flood case studies on the Central Apennines District.

Annalina Lombardi[1], Valentina Colaiuda[1,2], Marco Verdecchia[2] and Barbara Tomassetti[1]

[1]CETEMPS, Centre of Excellence University of L'Aquila, via Vetoio, 67010 Coppito (L'Aquila), Italy
[2]Department of Physical and Chemical Sciences, University of L'Aquila, via Vetoio, 67010 Coppito (L'Aquila) Italy

*Correspondence to*: Annalina Lombardi (annalina.lombardi@univaq.it)

**Abstract.** Flood and flash floods are complex events, depending on weather dynamics, basin physiographical characteristics, land use cover and water management. For this reason, prediction of such events usually deals with very accurate model tuning and validation, which is usually site-specific and based on climatological data, such as discharge time series or flood databases. In this work, we developed and tested two hydrological stress indices for flood detection in the Central Italy Apennine District: a heterogeneous geographical area, characterized by complex topography and medium-to-small catchment extension. Proposed indices are threshold-based and developed taking into account operational requirements of civil protection end-users. They are calibrated and tested through the application of signal theory, in order to overcome data scarcity over ungauged areas, as well as incomplete discharge time series. The validation has been carried out on a case study basis, through the use of flood reports from various source of information, as well as hydrometric level time series, which represent the actual hydrological quantity monitored by civil protection operators. Obtained results shows as the overall accuracy of flood prediction is greater than 0.8, with false alarm rates below 0.5 and probability of detection ranging from 0.51 to 0.80. A slight anticipation of peak occurrence was found, however, the time shift of indices signal peak it is strongly dependent on the presence of dams that regulates the flood propagation. Moreover, the different nature of the proposed indices suggests their application in a complementary way, as the index based on drained precipitation appears to be more sensible to rapid flood propagation in small tributaries, while the discharge-based index is particularly responsive to main channel dynamics.

## 1 Introduction

Floods are recognized among the most destructive natural hazards (Berz et al., 2001), affecting 21 mln people, globally, each year; unfortunately, this dramatic estimation is expected to rise up to 54 mln by 2030 (Lehman, 2015). So far, according to the data reported by MunichRe (2018), 2017 is considered the worst year in terms of overall losses caused by natural hazards.

It has also been long recognized as the increase in the frequencies of severe precipitation events represents a characteristic signature of observed climate changes at global scale; the intensification of the hydrological cycle due to the warming climate is projected to change river floods in both magnitude and frequency (Field et al. 2012, Blöschl et al., 2017). Kundzewicz and



Schellnhuber (2004) highlighted that about one-third of all reported events and one-third of the economic losses resulting from natural catastrophes are attributable to floods all over the world. Different works seek to analyse the impact of Climate Change Scenario on flood hazards in Europe, finding as several European countries will experience increasing flood risk in the future (Dankers and Feyen 2009; Feyen et al., 2012). Alfieri et al. (2015) shew a significant increase in the frequency of extreme events (i.e. larger than 100 %) in 21 out of 37 European countries, in the reference period 2006–2035, to be followed by a

further deterioration in the subsequent future. Blöschl et al. (2019) demonstrated clear regional patterns of both increasing and decreasing river flood discharges in the past five decades in Europe, attributable to changing climate. More specifically, the Mediterranean area is one of the climate system's most responsive hotspot to climate changes (Giorgi, 2006; Giorgi and Lionello, 2008). Indeed, 185 flood events were recorded in the Mediterranean countries between 1990 and 2006, being the number of cases affecting Spain, Italy and France 59% of the total. In the Italian Peninsula, these events caused 20 billion of

damages to buildings and infrastructures (Llasat et al., 2010). Mysiak et al. (2013) estimated that some 3.5 million people (6 % of the total Italian population) live in hydrogeological risk areas. Italy's history is characterized by many devastating floods, causing deaths and relevant economic losses and deep social and environmental impact. Given the high landscape variability, the complex topography and climatic variability, Italy is one of the most exposed countries to geomorphological risk. Meteorological patterns are frequently characterized by deep convective clouds, producing intense and localized rainfall,

rapidly developing in localized floods. Salvati et al. (2018) estimated that 441 flood events occurred over 420 Italian sites from 1965 to 2014, causing a total amount of 771 fatalities.

Considering the last two decades, Italy is the sixth country in the world for the number of victims caused by hydrogeological hazards and eighteenth in terms of economic losses (Eckstein et al., 2019). The European Parliament defined floods as *"the potential to cause fatalities, displacement of people, and damage to the environment, which can severely compromise economic*

*development"*. In the EU Directive 2007/60/CE concerning the "Assessment and management of flood risks", the realization of a flood risk map is foreseen over river basins with a significant potential risk of flooding (European Parliament, 2007). Prediction of flood events is therefore important to enhance mitigation strategies to face hydrological events. Since the 1970s, the hydrological forecast has improved (e.g. Jain et al., 2018, Ranit and Durge, 2018; Hapuarachchi et al., 2011); a comprehensive review of the different hydrological forecasting techniques is given in Teng (2017), where he finds that

empirical models are sufficiently suitable for post-event monitoring and analysis, while hydrodynamic models are indicated for dams and flash floods assessment. Eventually, simplified conceptual models are applicable for probabilistic flood risk assessment and multi-scenario modelling in well-defined channels. The data availability for the validation of hydrological models also influences the choice of the most suitable forecasting system (Jain et al., 2018, Cloke and Peppenberger, 2008). In general, precipitation indices are applied for flash floods prediction, since a negligible contribution of infiltration processes

is assumed for small catchments (Hurford et al. 2012; Ahn and Il Choi, 2013). Moreover, Alfieri et al. (2012) highlighted as precipitation-based indices are preferable over uninstrumented rivers. Schroeder et al. (2016) developed a flash flood severity index, universally applicable to all geographic locations, but many other authors had obtained better prediction scores by using runoff thresholds indices (Norbiato et al., 2009; Javelle et al., 2010; Raynaud et al., 2015; Alfieri et al. 2014), where thresholds





are chosen on a climatological basis, for a given return period. However, the application of such indices is limited to historically

monitored river segments, where a reference climatology is available. When historical runoff estimations are not available, validation is carried out on a case-study basis, if a reference flood hydrograph is available at station level (Nikolopoulos et al., 2013; Silvestro et al., 2015). Eventually, for validations of hydrological models assimilating rainfall estimation from remote sensing techniques, the reference flood hydrograph is obtained by forcing the hydrological model with rain gauges observations (Borga, 2002; Vieux and Bedient, 2004; Berenguer et al., 2005).

Many authors have recognized as an effective design of Early Warning Systems (EWSs) is a key element for fostering forecast skills and improve the resilience to natural hazards (Basha and Rus, 2007; Alfieri et al., 2011; Alfieri et al., 2012; Kundzewicz, 2012; Krzhizhanovskaya et al, 2012; Mysiak et al., 2013; Corral et al., 2019). In this framework, scientists in different fields have to deal with an effective development of new robust techniques and analyses. On the other hand, the achieved results need to be useful for the end, matching specific requirements. Horlick-Jones (1995) were the first to highlight the necessity of

structured collaboration between Civil Protection and scientist, in the framework of the United Nations International Decade of Natural Disaster Reduction.

In this work, we developed and validated two hydrological stress indices over Central Italy, currently used in the framework of the Agreement between the Centre of Excellence CETEMPS and the Abruzzo Regional Functional Centre, where the former was appointed as Competence Centre of the Italian Civil Protection and Abruzzo Region, as well. Due to its complex

topography, Central Apennines District (Central Italy, Figure 1) is characterized by both large and structured catchments (e.g. Tiber and Aterno-Pescara, see next section) and short ephemeral tributaries and torrents, which have a faster response to weather extremes and are more likely to be hit by a flash flood. Little information is available for those small catchments and hydrometric/discharge thresholds are, hence, difficult to define. As previously mentioned, precipitation thresholds are the most commonly used in this situation, together with climatologically-based regionalization procedures (e.g. Reed et al., 2007). The

lately discussed indices are meant to be used in a complementary way, having the advantage of being strongly user-oriented, as they are calibrated taking into account a correspondence between the issued civil protection alarm level and index threshold. The innovative nature of the presented hydrological stress indices lies in the definition of a unique index threshold, associated to an alarm state, which assumes the same value over each point of the drainage network reconstructed by the model. They have been conceived to be applied over an interregional domain, devoid of climatological hydro-meteorological time-series.

Before evaluating the performance of the hydrological forecast through the use of these indices, a procedure for their validation on past floods is to be defined, by assimilating observed meteorological data. The proposed evaluation procedure is designed to tackle with hydrological data scarceness and takes advantage from the signal theory processing methods.

The paper is organized as follows: a detailed description of the chosen hydrological model is given in section 2; in section 3, the flood forecasting system is described and the proposed hydrological stress indices are defined. Section 4 is devoted to give

a detailed description of the validation methods, while in section 5 the application of the proposed approach to several case studies is discussed.



## 2 Cetemps Hydrological Model

The Cetemps Hydrological Model (CHyM, hereafter) has been developed at Centre of Excellence Cetemps, since 2002

(Verdecchia et al., 2008b, Coppola et al., 2007). The original purpose was the development of an operational hydrological model for flood alert mapping (Tomassetti et al., 2005). However, the CHyM model has also been applied for climatological studies to investigate the effects of Climate Changes on the hydrological cycle (Coppola et al., 2014, Sangelantoni et al., 2019). CHyM is a fully distributed, physical-based hydrological model, where main hydrological processes are explicitly simulated by a physical-based numerical scheme.

An important characteristic of the model is the possibility to simulate the hydrological cycle over any geographical domain with any spatial resolution up to the DEM resolution (90 metres in the current version). Actually, the lower limit in choosing the spatial resolution deals with the validity of the numerical schemes used to simulate the hydrological processes (e.g.: the kinematic wave of shallow water, which used to solve the continuity equation, is considered a good approximation with a horizontal resolution of few hundreds of meters). In this section, the surface runoff calculation scheme is described in details;

other parameterizations, such as evapotranspiration, infiltration, melting and return flow, are described in Coppola et al. (2014).

### 2.1 Runoff

To simulate the surface runoff, the continuity equation for surface routing and channel flow is explicitly solved. The flow direction for each grid point is established following the minimum energy principle; therefore, the flow direction is assigned

to the adjacent grid-point located to the maximum downhill slope. The channel flow is computed according to the kinematic wave approximation of the shallow water equation (Lighthill and Whitham, 1955), where the continuity equation is expressed through the following simplified form:

$$\frac{\partial Q}{\partial x} + \frac{\partial A}{\partial t} = q \qquad (2.1.1)$$


where, $A$ is the flow cross-sectional area, $Q$ is the flow rate of water discharge (m³/s); q is the rate of lateral water inflow per unit of length, $t$ is the time, $x$ is the coordinate along the river path.

According to the shallow water approximation, the De Saint-Venant equation for the momentum conservation is expressed through the rating curve approximation:


$$Q = \alpha A^m \qquad (2.1.2)$$





where, α is the kinematic wave parameter, and *m* is the kinematic wave exponent, adimensional, which is assumed to be ≈ 1 for cylindrical river geometry. The kinematic wave parameter *α* has the dimension of a speed:


$$\alpha = \frac{S^{1/2} R^{2/3}}{n(\mu)}$$
(2.1.3)

where $S$ is the longitudinal bed slope of the flow element, $n$ is the Manning's roughness coefficient depending on the land use type $\mu$, $R$ is the hydraulic radius, considered as a linear function of the drained area $D$, according to the following formula:


$$R = \beta + \gamma D^\delta$$
(2.1.4)

where β, γ, and δ are empirical constants to tune in the calibration phase. If the hydraulic radius is expressed in meters and the drained area is expressed in $Km^2$, typical values of β, γ, and δ are respectively 0.0015, 0.35, and 0.33.

As for the surface flow outside the channel network, we assume that the surface water depth *y* is constant over each grid-point, therefore, the continuity equation assumes the following form:

$$\frac{\partial \varphi}{\partial x} + \frac{\partial y}{\partial t} = \xi$$
(2.1.5)

where $\varphi$ is flow rate over the longitudinal dimension (m²/s) of the grid point and the rate of water inflow per unit of area (m/s). The momentum equation has a linear relationship between the flow rate and the water depth, but Manning's roughness coefficient is increased by a factor $M_n$ as the water is assumed to flow with a lower speed. The value of $M_n$ is typically ≈4.5, but the optimal value is to be established during the calibration phase.

An arbitrary drained area threshold of 100 $Km^2$ is set to distinguish the overland flow from the channel flow, which is expected
to occur for drained areas wider than that threshold.

## 3 CHyM Flood stress indices for operational activities

Italian Legislative decree 02/01/2018 n. 1 defines, in art. 19, the role of the scientific community participating in the National Civil Protection Service, whose task is the development of products deriving from research and innovation activities aimed at
managing emergencies and risk preventing and forecasting. This study results from the need to identify useful and easy-to-understand tools for flood events prediction. Several flood events affecting the Italian peninsula in the last years have been analysed, in order to assess the possibility to predict a general-purpose alarm index to highlight the segments of drainage



network where critical stress are expected. The use of deterministic hydrological models for a hydrological forecast involves a series of critical points. First of all, the need to calibrate and validate models with very long time series of flow discharge
data. These data are not always available, in particular on small seasonal streams, usually not instrumented, but more prone to destructive flooding phenomena. Furthermore, there is significant uncertainty in river discharge estimations due to rating curve interpolation and extrapolation, the presence of unsteady flow conditions and the seasonal changes of the river roughness (Di Baldassarre and Montanari, 2009; Di Baldassarre and Claps, 2011). Secondly, it is really difficult to establish a flow discharge threshold value, beyond which the river can be considered under stress conditions; this value is site-specific and
refers to a certain river section, therefore, cannot be considered as general for the whole drainage network.

In the IPCC SREX report (Field et al., 2012), floods are defined as: "*the overflowing of the normal confines of a stream or other body of water or the accumulation of water over areas that are not normally submerged. Floods include river (fluvial) floods, flash floods, urban floods, pluvial floods, sewer floods, coastal floods, and glacial lake outburst floods.*".

Precipitation intensity, duration, amount and timing are the principal mechanisms affecting a flood event. Moreover, the
relationship between the rainfall and drainage network response is complex (Bates et al., 2008; Kundzewicz et al., 2012) and sensible to rain spatial distribution. In large river basins, for example, river floods are generated by intense and enduring rain while short-duration, highly intense rainfall is expected to determine floods in small basins. In Chen et al., 2010 is highlighted different flooding drivers exist; the main ones are: pluvial flood, due to the limited capacity of a drainage system and fluvial flood, caused by deluges from the river channel. The fluvial flood events considerably differ from pluvial (rainfall) flood
events both in spatio-temporal scale including its magnitude. The fluvial events usually occur for the duration of days or even weeks with widespread damages in the floodplains of the river system. On the other hand, pluvial flooding hardly ever happens for more than one day duration with an influence on local regions (Chen et al., 2010; Patra et al., 2016; Apel et al., 2016). Frequently, pluvial flooding happens together with fluvial flooding but it is important to highlight the influences of various flooding drivers (Ashley et al., 2005; Balmforth et al., 2006). Those flood scenarios were derived, for example, by adding
rainstorms to the fluvial flood events and this condition is easily found when we consider Italian river basins with a size even up to a few tens of km. In this case, fluvial and pluvial floods are combined and are sufficient from a few days to a few hours of intense rainfall depending on the considered basin. For this reason, we developed two different indices linked to the different flooding sources: CHyM Alert Index (CAI) a pluvial flood index, related to the limited capacity of a drainage systems, and Best Discharge-based Drainage (BDD) a fluvial flood index, related to deluges from river channels. The idea of hydrological
stress indices comes from the collaboration with Civil Protection. These indices and the associated stress thresholds are general; the signal of the hydrological forecast is easy and quick to understand. The hydrological stress indices use the quantity of drained water and the geomorphological characteristics of the different rebuilder basins. Daily operational activities for flood alert mapping are carried out using CHyM model. To encompass the whole Italian peninsula, the hydrological forecast is divided into 7 sub-domains at different spatial resolutions (Taraglio et al., 2019; Colaiuda et al., 2020). The spatial resolution
depends on the size of the simulated basins.



## 3.1 CHyM Alert Index

The CHyM Alert Index (CAI) has been long used for the operational activities of flood alert mapping in Central Italy. CAI is calculated as a function of the rainfall drained by each elementary cell of the simulated geographical domain. More specifically, the index is associated to each grid-point, being the ratio between the total drained precipitation and total drained area in the upstream basin, up to the specific grid-point. The proposed definition of the hydrological stress index has also a simple physical interpretation: it represents the average precipitation drained by each cell, considering the rain falling over the whole upstream basin of the selected cell, during a time interval corresponding to the mean concentration time. A first version of the CAI index is described and tested in Tomassetti et al. (2005) and Verdecchia et al. (2008a); in its initial formulation, the mean concentration time of the upstream basin was considered as a fixed term (36 or 48 hours, depending on the basin dimension). An updated version is presented in this work, where the average corrivation $t_c$ is explicitly calculated from each drainage path $k$, down to the considered grid-point of coordinate $i,j$

$$\overline{t_c^{i,j}} = \sum_{k=1}^{N} \frac{t_{k \to ij}^{i,j}}{N} \tag{3.1.1}$$

where $N$ indicates the total possible flowing paths. The updated formula of the CAI index is then the following:

$$CAI = \frac{\int_{UP} \int_{t-\Delta t}^{t} P(t,s) dt ds}{\int_{UP} ds} \tag{3.1.2}$$

being $P$ the precipitation available for the runoff. The integral over the space $s$ is calculated considering the whole upstream basin of the selected cell. For the stress state identification, three different thresholds have been defined, after carrying out empirical tests: each threshold has been adequately chosen to qualitatively match the three different civil protection state of hydrological criticality, as defined by the Head of Civil Protection Department (2016):

1. Ordinary Stress: 30 mm/day;
2. Moderate Stress: 60 mm/day;
3. High Stress: 110 mm/day.

The definition of each hydrogeological criticality level (and related colour-codes) is summarized in Table 1.

## 3.2 BDD Best Discharge-based Drainage index

The BDD index is linked to the CHyM predicted discharge and is calculated, for each grid-cell of the drainage network, according to the following formula:



$$BDD(t) = \frac{Q(t)}{R^2} \tag{3.2.1}$$

where $Q$ is the discharge predicted at time $t$ and $R$ is the hydraulic radius of the selected elementary cell, calculated as a linear function of the upstream basin (see equation 2.1.4). BDD stress thresholds have been chosen following the same meaning of the CAI thresholds, in order to match the three relevant hydrological criticality levels:

1. Ordinary: 3mm/h; 6mm/h;
2. Moderate stress: 6 mm/h;

3. High stress: 11 mm/h.

## 4 Materials and methods

Floods are complex events and data collection is not an easy task to achieve in this matter. The Italian government introduced the "Cadastre of Events", in response to Directive 2000/60/CE, a registry where relevant hydro-meteorological events are

listed and associated to a heterogeneous database of different territorial data, organised in geo-referred layers (e.g. flood time, localization and damages). Data sources are not necessarily objective measurements: collections may contain official Civil Protection reports and press releases, as well as other reports from local authorities. The official Hydrogeological Catastrophes GIS archive is available online at: http://sici.irpi.cnr.it. However, the database update was concluded in 2000: after this date, only few Italian Regions had moved to an alternative way of data collection, mainly represented by regional databases with

different data structure and classification, freely or restrictively accessible for external users. Considering the lack of official updated databases in the studied area, a huge, but necessary, effort was carried out to collect all available territorial data for the selected case studies, following the approach of the "Cadastre of Events", in order to create an Own Data Base (ODB) with territorial, geo-referred information. Collected information in ODB were used as reference data for the indices validation process, discussed in the following section.

ODB was filled by searching and classifying the following heterogeneous data about the considered flood events:

- official Civil Protection Event Reports, issued by regional Functional Centres or environmental agencies;
- COPERNICUS Emergency Management Service;
- POLARIS database by CNR-IRPI;
- data from the AVI Project;

- press releases;
- photographic documentation from social media (e.g. YouTube, YouReporter, etc…), reporting major rainfall events, floods and landslides causing direct human consequences and damages in the investigated period;
- available hydrometric level time series and thresholds, where updated;



The above listed information were not all available for the same case study (CS), for this reason, a summary of the found
validation material for each event is reported in Table 2. Moreover, in order to provide an overview of the data collection
geographical distribution, the same information listed in Table 2 have been geo-referend and shown in the map of Figures 2,
3 and 4.

Besides the territorial information, other hydrological data were used for the validation process. The Italian Prime Minister
Decree (DPCM), issued on 27 February 2004 and concerning the "Operating concepts for functional management of national
and regional alert system during flooding and landslide events for civil protection activities purposes", establishes the Regional
Functional Centres to acquire and collect real-time data from monitoring networks. Hydrometric levels are identified as the
quantities to be monitored in order to assign the critical level for, at least, moderate and high hydraulic risk to each warning
area, through the definition of thresholds. Article 5 of the same Decree define as the real-time validation of prediction systems
is made through the monitoring of moderate and high hydrometric level thresholds exceedances, for the main river channels.
Secondary drainage network with drained area less than 400 km$^2$ are not included in this kind of validation.

The definition of water level critical thresholds (Italian Laws no. 59/2004, Fassi et al. 2008), is carried out for each Italian
Region by local Civil Protection Authorities (Regional Functional Centres) at station level (Fassi et al., 2008; Brandolini et
al., 2012; Mysiak et al., 2013). A colour code is then assigned to each hydrometric threshold (see details in table 1), indicating
four different alarm level, corresponding to specific hydraulic risk management actions, activated at the institutional level
(Italian Legislative Decree no. 01/2018). However, as recognized by the Italian Institute for Environmental Protection and
Research (ISPRA), the hydrometric level is a strongly non-stationary variable, as it is influenced by the riverbed erosion and
deposition processes (Braca et al., 2013). The hydrometric zero needs to recalibrated, establishing an updating frequency
adequate for the river flow regime and local hydrogeological factors. Moreover, the calibration should be carried out after
flood occurrences, when the riverbed shape is significantly modified. Then, the hydrometric thresholds need to be revised
correspondingly. After the application of the Italian Law no. 183/1989, the management of the gauge's network and data
collection is devoted to the Regional authorities. Even though a territorial approach is useful for a rapid response to risk
scenarios, competences fragmentation among different entities had caused inhomogeneities on hydro-meteorological data
availability and quality (e.g. rating curve updates, historical hydro-meteorological data time series, hydrometric threshold
availability for all stations, etc...) with significant differences among the twenty Italian regions.

For all the aforementioned reasons, a deterministic hydrological flood prediction validation over a wide, interregional area can
be challenging or not universally applicable, due to missing or obsolete information. Moreover, the discharge computation in
hydrological models is affected by systematic biases when the hydrological network is exploited for hydropower production,
irrigation or industrial and domestic usage: in most cases, data about water uptake are scanty or incomplete, as they are
collected by a variety of public and private actors and difficult to obtain. Another common issue for the spatial validation deals
with thresholds inference on ungauged areas. Alfieri et al. (2019) highlighted that floods and flash floods usually occur in
ungauged catchments: for those situations, post-event survey reports represent the only source of information. Besides, even
if present, gauges data may be unavailable during a severe event or damaged by the flood.





The hydrological stress indices validation was first assessed through a qualitative approach, by selecting the strongest recorded signal of upcoming severe events from the hydrometric level time series and verifying the actual occurrence of floods in the

areas where they were forecasted. To this aim, hydrological stress indices maps are compared with ODB geo-referred maps. In addition, an objective analysis is carried out by applying both statistical dichotomous and continuous scores.

### 4.1 Statistical dichotomous analysis

Primarily, the indices grid map was spatially co-located with the hydrometers position by choosing the nearest grid-point to

the station geographical coordinates after verifying the correspondence between the grid-point upstream drained area calculated by the CHyM model, with the real value declared in the official station registry (where available). As for the time co-location, both water level and indices time series are hourly, and it might appear straightforward to investigate the potential thresholds exceedances by comparing the same time step. However, during a flood wave, it is not infrequent to have water level data corrupted by measurement errors during the flood wave transition (i.e.: a solid surface stationing for a certain period

under the hydrometric sensor). For this reason, the time location is carried out by associating a mobile interval of three hours (the target time step ± 1) of observations to each index time step. The choice of this confidence interval is arbitrary, although it based on the authors' experience. The contingency table was then built, for each station point and for each index, considering the match between the co-located moderate hydrometric threshold exceedances (THR 2 in table 1) and the moderate indices threshold exceedances. Differently from water level thresholds, CAI and BDD indices thresholds have the same value for the

whole grid points of drainage network. These numerical thresholds are respectively 6 mm/hour for BDD and 60 mm/hour for CAI: the choice of these values is justified *a-posteriori* considering the performances of the proposed indices for different analysed severe events, during 10 years of operational activity (Colaiuda et al., 2020). Under the most natural conditions and with continuous updating of the hydrometric thresholds depending on the morphodynamic variability of the basin, the proposed threshold levels for BDD and CAI should appear to be very close to the water level threshold for the specific site.

The dichotomous scores include the accuracy (A), the probability of detection (POD), the false alarm ratio (FAR). To build such a table, a flood event is considered as an observed "yes/no" event if the water level exceeds/does-not-exceed the empirical threshold; a flood event is an estimated "yes/no" event if the estimated index exceeds/does-not-exceed the BDD and CAI thresholds (Table 3). *A*, *POD* and *FAR* scored are defined as follows:


$$A = \frac{H+CN}{H+M+FA+CN} \tag{4.1.1}$$

$$POD = \frac{H}{H+M} \tag{4.1.2}$$

$$FAR = \frac{FA}{H+FA} \tag{4.1.3}$$






The calibration of the indices thresholds was chosen in order to maximize the hit rate *H*, according to Alfieri et al. (2019). All listed scores ranges from 0 to 1, where 1 is the optimal value for *A* and *POD*, while 0 indicates the best possible score for FAR.

## 4.2 Catch Rate

The Catch Rate (CR) was estimated for each index, in order to investigate the effectiveness in detecting or missing correct flood warnings. To this aim, the orange (moderate) hydrometric level threshold exceedances (THR2) were chosen as a term of comparison with the corresponding moderate CAI and BDD indices thresholds. A match occurs when the hydrometric THR2 is exceeded and the moderate index threshold is exceeded, or, when the hydrometric THR2 is not exceeded and the moderate index threshold is not exceeded, within a 24 hours time range. A Boolean value 0/1 is then assigned when a match occurs. CR

is then calculated as the ratio between the number of correct matches found and the total number of analysed stations *N*:

$$CR = \sum_{i=1}^{N} \frac{1}{N} CESA_i \tag{4.2.1}$$

Where the acronym *"CESA"* stands for Correct Estimated State of Alert for the *i*-sensor, which assumes value "1" when
estimation matches observation, and "0" when that match does not occur.

## 4.3 Time Peak Analysis

In order to further evaluate the timing accuracy of the BDD and CAI indices, all the available observed water level time series were compared to the indices time series. Because of the comparison between two different physical quantities, the chosen
statistical scores are typically used for signal studies. The first statistical analysis was made through the calculation of the Lag Time Peak (LTP), in order to investigate the simultaneity of occurrence between the water level peak and the indices peak. According to the Italian Prime Minister Directive concerning "Operational guidelines for emergency management", issued on 3 December 2008, a lag time of "a few hours" (less than 12 hours) is estimated to be between an event occurrence and the activation of the Civil Protection Coordination Unit. In light of the above, we established that an adequate lag time peak for
flood prediction should not exceed 3 hours. According to other authors (see, as an example Rabuffetti et al., 2008), the Relative Lag Time Peak (RLTP), defined as the ratio between LTP and the average time of concentration of the upstream basin.





## 4.4 Correlation Time Delay (CTD)

The cross correlation (CC) is typically used in the signal theory (Rabiner and Gold, 1975; Rabiner and Schafer, 1978; Benesty

et al., 2004), for the assessment of the similarity between two signals. Given two discrete series $x(t)$ and $y(t)$, each one of N components, the cross correlation is calculated as the dot product of the series:

$$CC = \sum_{i=1}^{N} x(t_i)y(t_i) \qquad (4.4.1)$$

The same product can be calculated, shifting the two signals of a time lag $L$:

$$CC(L) = \sum_{i=1}^{N} x(t_i)y(t_i + L) \qquad (4.4.2)$$

The Correlation Time Delay (CTD) is then defined as the value of time lag $L$ that maximizes the previous product.


$$CTD = \max_{L \in R} CC(L) \qquad (4.4.3)$$

CTD represents an estimation of time shift between two series; therefore, we found this score to be suitable to measure the effectiveness of the signal given by the hydrological stress indices.


## 4.5 Derivate Dynamic Time Warping analysis

The Dynamic Time Warping (DTW, Berndt and Clifford, 1994; Keogh and Ratanamahatana, 2005; Maier-Gerber et al., 2019 and Di Muzio et al., 2019) allows to stress (or compress) two-time series to achieve a reasonable fit between them. The idea of the method is that the similarity between two sequences can be estimated by "warping" the time axis of one (or both)

sequences, in order to achieve a better alignment. Although DTW has been successfully used in many domains, it may lead to obtaining wrong results; as an example, the technique may fail in finding the optimal alignment because a feature (i.e. peak or local minimum) in one sequence is higher or lower than its corresponding feature in the other sequence.

To overcome this problem, Keogh and Pazzani (2001) proposed the computation of warping using the local derivative of the time series to be compared, and called this algorithm Derivative Dynamic Time Warping (DDTW).

The numerical procedure for the DTW calculation can be summarized as follows: given two discrete series $x(i)$ and $y(j)$ of N and $M$ components respectively, an N-by-M matrix is built. An element $V(i,j)$ contains the Euclidean distance between the i-th element of the first sequence and j-th element of the second sequence. For this matrix, a "warping" path $W$ is defined as a contiguous set of $L$ matrix elements, and the measure of misalignment $d$ for the path $W$ is given by:





$$d(W) = \frac{\sum_{i,j} V(i,j)}{\frac{1}{2}L(L-1)} \qquad (4.5.1)$$

where the sum in the numerator is carried out over all the elements belonging to the warping path *W*. The denominator is used to normalize different length sequences. The DTW index is then calculated as the minimum value of *d(W)*, considering all the possible path *W*.


$$DTW = \min_{W} \ d(W) \qquad (4.5.2)$$

For instance, if the two considered sequences are aligned and have the same number of components (*N=M*), the optimal path will be the *N* diagonal elements of matrix *V*.


The DDTW (Figure 5) algorithm implementation replaces the data time series with their first derivative and the Euclidean distance is measured on them. The first derivative has been calculated for each time serie as follows

$$D(x[i]) = \frac{(x[i]-x[i-1])+((x[i+1]-x[i-1])/2}{2} \qquad (4.5.3)$$

**5 Results and discussion**

In this section, the analysis of a meteo-hydrological event occurred in Central Italy on 11-13 November 2013 is proposed. The 3-days event was characterized by intense precipitation, involving the whole Central Apennines ridge and three different regions, progressively affecting the Adriatic side of the central part of Italy, moving from North to South. In order to better organize our analysis, the event was divided into three different case studies, related to three different regions involved: Umbria 400  (CS01), Marche (CS02), and Abruzzo (CS03). The CHyM model simulations were set to three different geographical domains, as shown in Figure 6. The event was very intense and caused many damages and few fatalities in all regions: an overview of the phenomenon is reported in Table 2, where relevant information about observed effects and sources of information are provided. Details of links pointing to each source used to organize ODB are provided in the supplementary materials, where all hit municipalities and affected rivers are also listed.


**5.1 Study Area Description**

The study area covers the Central Apennines District (Figure 1), with an extension of 42 506 km$^2$ and about 8 million inhabitants.  The northern part, which encloses the upstream basin of the Tiber river, from the confluence with the Nera river, is characterized by a less dense draining network with respect to the lower part of the basin. This area has an extensive



hydrography, characterized by both perennial rivers, constantly fed by groundwater, and seasonal streams, which are activated only in rainy periods. Moreover, plenty of artificial reservoirs and hilly ponds uptake surface runoff water. The Adriatic slope is located over the central part of the district, extending from upper Marche Region (Potenza River) to the southern part of Abruzzo Region (Sangro River). The lower path of the Tiber river is also part of this area, together with the tributaries on the left bank, from Nera to Aniene rivers.

This area is affected by inundations along major rivers, as well as flash floods in torrents and minor streams, especially on the heels of the ridge, where high-intensity rainstorms cause lowland floodings.

Most portion of the drainage network is characterized by significant water storage (with a quite constant spring flow rate during the year) and marked by hydroelectric power plants, built since the last century (Tiber Basin Authority, 2010).

The peak discharge variation depends on the storage type: generally, the effect of a reservoir to flood control results from a 420 combination of regulated and unregulated storage (Volpi et al., 2018). The former, used in the analysed area, is less efficient in flood-peak reduction than regulated storage as it begins filling even before it is needed. Moreover, the effect of a flood control reservoir depends on the combination of off-stream or on-stream detention ponds: for example, according to Ravazzani et al. (2014). Dams and reservoirs play an important role during flood events (Rodda, 2011; Kundzewicz et al., 2014; Ayalew et al., 2017; Habets et al., 2018): this role is not always favourable; they adversely affect the extent of an inundation due to 425 dike breaches, blockage of bridges and culverts by debris. Anyway, weak coordination between different actors involved in water resources management may significantly affect flood dynamics. In multi-purpose reservoirs, competing interests represent a key issue in flood regulation: irrigation, hydropower generation and flood control generally compete, even when the reservoir is owned by a single country or agency. This conflict of interests is heightened, when the basin is interregional, as in the case of Central Apennines District. For those reasons, the WMO (2009) recommends to carefully evaluate the flood 430 timing and dynamics.

## 5.2 Synoptic Analysis

On November 11, 2013, a large synoptic-scale meteorological system originated from the Atlantic Ocean and moved into the Mediterranean area. In particular, the cold air coming in from the Rhone door has induced rapid cyclogenesis on the Genoa 435 Gulf. The barometric minimum moved southward along the Italian Peninsula and reached the Tyrrenian Sea on November 12. The persistence of the occluded front over Italy caused heavy and long precipitation, initially affecting northern regions and, progressively, central and southern areas, as the minimum moved toward the Lybian coasts, on November 13. The precipitation was widespread, with a huge amount. According to the event reports from Regional Civil Protection authorities, registered precipitation amounts were almost than 300 mm/72 hours in several areas, mainly located along the Apennines ridge, between 440 Marche and Umbria regions (Figure 7).



### 5.3 Case Studies Analysis

Hydrological simulations were carried out over a geographical domain larger than the areas where floods were actually observed, in order to verify the absence of predicted hydrological stress conditions in those areas where hydrological

criticalities did not occur. Hydrological simulation was set by using a spin-up time of 120 hours for all case studies, before the day of the hydrological event. The selected case studies affected different regions of Central Italy characterized by catchments of different sizes and geomorphological characteristics, allowing the evaluation of indices feasibility in heterogeneous domains. Spatial and temporal characteristics of the hydrological simulations are reported in Table 4.

As discussed in section 4, the ODB information about case studies were geo-referenced on a Google Earth map (Figure 2, 3,

4). The cyan waves symbols indicates reported inundation and the pinpoints show the hydrometers displacement: the colour assigned to each pinpoint highlights the observed state of alert, namely, the hydrometric threshold exceedances (see Table 1 for further details). In the same map, the drainage network is represented by blue lines; white lines indicates alert zone boundaries, defined by Civil Protection, reddish areas encompass the administrative boundaries of the main affected municipalities (i.e. where a flood was reported), while the small cyan triangles highlight the main water reservoirs located

inside the domain. In Figure 2 and 3, red rectangles represents the flood involved area published on Copernicus Emergency Management Service Platform (EMS Rapid Mapping Activations (EMSR060): https://emergency.copernicus.eu/mapping/list-of-components/EMSR060).

### 5.3.1 Case study 1: Umbria Region

From November 11 to 12, 2013, a severe weather event hit the Umbria Region. The event mainly concentrated over the North-eastern part of the region, along the administrative boundary with Marche Region. According to the data provided by the official hydro-meteorological monitoring network, precipitation was persistent and intense, resulting in exceptional amounts, up to 440 mm in the Castelluccio di Norcia station and 330 mm in Gualdo Tadino, in 72 hours (see Figura 7). Floodings affected main rivers, as well as small catchments (river outlet highlighted in Figure 2), such as the Tiber, the upper Chiascio,

and the Topino basins. In particular, the flooding on the Sentino river, flowing along the boundary with the Marche region, caused damages on 12 residential buildings and temporarily isolated the Branca Hospital, due to considerable roads and bridges disruption. All municipalities of the Apennines ridge registered damages. A flood wave occurred over the Nera river and the Corno tributary. According to the Civil Protection official report, Montedoglio and Corbara dams played a crucial role in the flood wave lamination and phase shifting in the Tiber and Casanuova rivers and on the Chiascio river, respectively (Figure 5,

the bigger cyan triangles). For this first Case study (CS01, Figure 6), the main characteristics of hydrological simulation are reported in Table 4. The hydrological model has been forced with observed precipitation data from almost 370 rain gauges, located in the geographical domain.

The CAI and BDD indices maps obtained for CS01 are shown in Figure 8. Hydrological stress indices are computed at hourly time steps. However, the map refers to a 24 hours time interval, where the maximum daily value of the index is assigned to



each grid-point. In other words, the map gives an idea of the maximum stress conditions that may occur in the whole day. Moreover, the actual drainage network is more dense in the highlighted catchments, however, we decided to plot only grid-points having a drained area larger than 15 km$^2$, in order to improve the map visualization and interpretation.

A qualitative comparison of Figures 2 and 8 allows to identify similarities in the hydrological stress spatial distribution and observed inundations. The higher CAI and BDD stress degree are mainly given in the north-eastern side of drainage network,
from the upper Umbria regional boundary and along the slope exposed to the Adriatic side. All the reported damages and orange/red hydrometric levels are observed in the same area. The western side of Umbria was not significantly affected by the event and no relevant stress degree is given by indices.

The CAI index overestimates the hydrological stress extension in the south-eastern part of the region, near the boundary with the Lazio region. Moreover, a difference between the two indices needs to be highlighted: the BDD index stress degree is
lower in the minor drainage network, and relevant in the main river channel. This effect is due to the different nature of the indices and the different physical quantities considered in their calculation: the CAI index is directly linked to the precipitation rate, resulting in higher responsiveness in the smallest river channels, where the high peak precipitation-driven flood is the predominant mechanism for inundations. On the other hand, the BDD index calculation is based on the discharge value and responsive to runoff-driven floods, resulting from the combination of different hydrological processes, such as rainfall-runoff,
infiltration, soil moisture, and melting.

Besides the spatial assessment, timing analysis is given though a heterogeneous comparison between hydrometric level time series and indices time series (Figure 9), for six relevant hydrometric stations located in the upper part of Umbria region, where the orange hydrometric threshold was exceeded. The represented quantities, as well as the associated thresholds, are normalized. Indices increasing and decreasing rates show a similar behaviour with the hydrometric level trend, and maxima
occurrence is concomitant in Tiber and Topino river stations and slightly anticipated (from 3 to 6 hours) by the indices in Chiascio, and Nera rivers.

As stated at the beginning of this paragraph, artificial water storage and lamination played an important role in the flood wave management in this area. Flood abatement is achieved by detaining and later releasing a portion of the peak flood flow (WMO, 2009) and different kinds of reservoirs have been found to cause different release dynamics.
Focusing our analysis on the Pianello station (Figure 9d), downstream the Casanuova dam, the indices peak shifting may be due to lamination of flood wave by the on-stream water storage system. The presence of the Montedoglio dam, upstream Tevere-Santa Lucia and Tevere-Petrantonio stations (Figure 9 a and b) does not produce the same effect and peaks timing appears concomitant with the hydrometric level peak. Although the dam has retained almost all of the inflows from the upstream basin (up to 13 November at 12UTC), its off-stream position allowed the regulation of the intensity of the flood,
rather than its timing (Figure 9a, b)). The increasing hydrometric profile after 13 November 2013, 12 UTC, indicates the artificial release after the end of the event.





Despite the presence of large and small detection storages affecting the indices accuracy, the average CR value, calculated over 22 hydrometric stations is 0.86 for BDD and 0.77 for CAI. This result supports the qualitative analysis discussion, where a spatial correspondence between the indices map and the ODB map was observed.

Threshold exceedances hourly match have been calculated over the same 22 hydrometric stations, belonging to 7 different catchments, for a total amount of 2574 hours analysed (120 hours per station); resulting scores are summarized in table 5 (scores for each station are not shown). The accuracy of the prediction is above 0.8 for both indices, while the POD is around 0.7 for BDD and 0.5 for CAI. False Alarm Rate is around 0.45 in both cases, however, when the flood dynamics are artificially regulated, the misalignment between observed peaks and simulated peaks, analysed in Figure 9 discussion, leads to increasing

values of FAR. As for the timing analysis, resulting LTP is <1 hour for BDD and about -7 hours for CAI index. In general, all timing scores resulted to be better for BDD than CAI index, with a slight tendency to anticipate the peak values.

### 5.3.2 Case study 2: Marche Region

The official report of the Marche Region Civil Protection described the occurrence of many hydrological criticalities over the

whole region, especially in the inner areas, along the Apennines ridge, where rainfall maxima were registered. Precipitation peaks were up to 487 mm in 72 hours in Pintura di Bolognano rain gauge (South-East of the region) and 200 mm in Conca_1 (North-East of the region) (Figure 7). Floods affected many rivers in the upper Marche: critical situation over the Metauro basin was recorded for the upstream tributaries (Candigliano, Bosso, and Burano rivers). Other floods were also recorded in the Cesano, Misa, and Arzilla basins, as well as the Chienti river, where a flood wave propagated, starting from the upper

tributaries Fiastrone and Fiastra. On the Foglia basin, the Mercatale dam laminated part of the flood since early hours of November 11, until the afternoon, when its accumulation capacity finished. The upper part of the Potenza, Tenna and Tronto basins was also affected by floods.

A qualitative analysis of hydrological stress spatial distribution by comparing ODB data (Figure 3) and indices maps (Figure 10) was carried out, in order to identify a geographical match between recorded inundation and simulated hydrological stress.

For this first Case study (CS02, Figure 6), the main characteristics of hydrological simulation are summarized in Table 4.

The hydrological model has been forced with almost 138 rain gauges data, used to rebuild the precipitation field. Main affected municipalities lay on the piedmont areas of the Apennines' ridge (the lower part of Figure 3), where maxima precipitation was registered. However, the flood wave originating from those areas propagated downhill, toward the Adriatic Sea, affecting all the river systems, where damages, inundations and hydrological level criticality threshold were exceeded. The CAI index map

(Figure 10b)) identifies almost each grid-point of the drainage network with the highest stress degree, while The BDD map identifies maximum stress over the main rivers (Figure 10a). The difference between the two indices different behavior is due to the same mechanism described for CS01.

In Marche region, many dams affect the natural river flow. The normalized indices and hydrometric levels profiles are shown in Figure 11: the effect of Mercatale dam lamination caused a progressively larger hydrometric peak shifting along the Foglia

river (Figures 11c and 11d). The precipitation resulted in high supplies to the reservoirs of the Metauro, Chienti and Tronto




basins where it was necessary to retain part of the inlet flow during the event. Where the storage capacity allowed to manage the amount of precipitated rain, a rolling service was carried out. Maximum peak shifting is shown in Figure 11f, for the Fiastrone station. A first hydrometric peak is slightly postponed by indices, while a same-magnitude secondary peak is weakly detected. A relevant impact on Fiastrone station is determined by the presence of unregulated on-stream storage from the

Fiastra lake, which is the largest hydroelectric basin in the Marche region. Other time series shown, in Figure 11, are characterized by a synchronous peak of indices and hydrometric level. Dichotomous and continuous analysis scores for CS02 are shown in Table 5: for this case study, 28 stations time series have been analysed, covering 13 different basins. In this case, an Accuracy of 0.8 is reported for both indices, being the POD around 0.70 for BDD and 0.55 for CAI. A FAR score of 0.37 for BDD and 0.43 for CAI has been calculated. LTP is less than one hour for BDD and -4 hours for CAI index, resulting in a

RLTP of 0.05 for BDD and -0.56 for CAI index. CTD is significantly lower in BDD respect to CAI index, with values of -1.4 and -5.6, respectively. The DDTW results to be 0.04 for BDD and 0.06 for CAI index. Indices response on dichotomous scores (CR) is found to be similar, however, timing scores are quite different, resulting in slight anticipation of peak values in the CAI index. Worst scores were obtained on the Fiastrone river, heavily impacted by the Fiastra dam lamination. The effect on indices timing is relevant since the flood wave is simulated to occur 22 hours in advance for the BDD and 32 for the CAI. For

this station, despite values for FAR ranging from 0.56 for CAI to 0.56 for BDD, the POD scores are 0.94 and 0.66, respectively.

### 5.3.3 Case study 3: Abruzzo Region

In the Northern and Central part of Abruzzo region, a precipitation amounts up to 400 mm in 72 hours were recorded in Pretara rain gauge and 280 mm in Castel del Monte, over the inner, mountainous area (Figure 7).

Main affected rivers (from North to South) were Salinello, Tordino, Vomano, Piomba, Saline (included its tributaries Fino and Tavo), and Pescara, the latter one being the widest catchment of the region with a drained area of 3190 km$^2$. All aforementioned catchments, as well as the whole Abruzzo region territory, is disseminated by plenty of water withdrawals: according to ISPRA (2018), 14 relevant dams retain a total amount 370.38 mln m$^3$ of water from the drainage network. An undefined number of minor withdrawals is still under census by the local authorities (Abruzzo Region Deliberation no. 435/2016) and the total

magnitude of the water uptake is still difficult to asses.

Flood event for this case study affected the road networks, industrial settlements, scattered houses located along the river paths and in depressed areas. Flood waves also occurred in the southern part of the region although they did not have significant effects. Main setting of CS03 (Figure 6) are summarized in Table 4. The hydrological model assimilated data from 135 rain gauges from the official network. The comparison between indices 24 hours maps (Figure 12) and ODB observation spatial

distribution (Figure 4) reveals a correspondence between the most damaged area, involving all river systems North of Pescara river, and the highest hydrological stress, highlighted by the indices reddish colours. However, the CAI index map also shows high hydrological stress in those southern watersheds, where inundations are not reported. On the other hand, according to the BDD map, no relevant stress is detected in this area. Moreover, the smallest tributaries were not highlighted by the latter index, coherently with CS01 and CS02 findings. In Figure 13, six relevant normalized time series of hydrometric levels and indices





are reported. Tordino, Pescara, and Vomano stations show peak shifting, due to the presence of many dams along the rivers
      path. For example, the Aterno-Pescara catchment hosts at least 7 different dams, concentrated in a drained area of barely 3100
      $km^2$. The induced flood shift may be even in the order of more than one day (e.g. 30 hours delay in Pescara a Villareia station,
      Figure 11f). Picciano and Fino are small watersheds, not impacted by human activity in terms of water uptake: in this case
      timing scores assumes lower values (LTP and CTD about 2 hours for BDD and -1 hour for CAI, with RLTP of -0.29 and -
0.14, respectively). DDTW results to be 0.09 for BDD and 0.11 for CAI.

      Among all the sensors analyzed, the indices reported the right state of criticality for about 78% of them (CR scores, Table 5).
      The timing analysis is given for 26 hydrometric time series, located over 16 different river basins. The overall Accuracy for
      CS03 is more than 0.9 for both indices, with a higher POD for BDD (0.81), than for CAI (0.72). However, the latter shows a
      slightly lower FAR. LTP is about -8 hours for BDD and -12 for CAI, resulting in RLTP of -1.1 for BDD and -1.6 for CAI. The
Correlation Time Delay is lower for BDD than for CAI (-4.2 and -7.5, respectively), while the DDTW is very low for both the
      proposed alarm indices, as it is 0.09 for BDD e 0.1 for CAI. Obtained results suggest that almost all floodings were predicted
      by both indices, even if the timing analysis reveals slight anticipation, probably due to the artificial water management effect.

## 6 Conclusions

      This work focused on flood prediction through the application of end users-oriented indices, able to identify segments of
drainage network susceptible to flood. Several hydro-meteorological severe events, collected during the CETEMPS
      operational activity, were used to calibrate two hydrological indices thresholds, starting from the calculation schemes of the
      CHyM model. The use of deterministic models for hydrological forecast involves a series of critical points. First of all, the
      need to calibrate and validate the model outputs with a very long time series of hydrological quantities, mainly represented by
      discharge data. However, these data are not always available, in particular, on small seasonal streams that are not remotely
monitored, but frequently hit by destructive flooding phenomena. Since floodings are complex events, depending on several
      processes, it is not straightforward to establish a flow discharge threshold value, beyond which the river can be considered
      susceptible of flood. For this reason, many developed hydrological thresholds are site-specific and not generally applicable
      over different areas, other than the river sections where they have been calibrated. The two proposed CAI and BDD indices
      were validated on a case study basis, through the analysis of an extreme weather event affecting Central Italy on 11– 13
November 2013. The 3-days event was simulated by the Cetemps Hydrological Model, forced with observed raingauges data,
      over three different geographical domains encompassing Umbria and Tiber basin, Marche and Abruzzo regions. Indices
      formulations followed two different approaches: the BDD index is based on the ratio between the computed (natural) discharge
      and the square of hydraulic radius, while the CAI index is more empirical, representing the amount of the precipitation drained
      by each grid-point of the drainage network, in a time interval corresponding to the mean concentration time of the upstream
area. Three thresholds have been set for each index and calibrated in order to obtain a qualitative correspondence between the
      indices and the hydrometric thresholds exceedances, defined by each Regional Civil Protection Functional Centre. A colour-





code, similar to those used for the hydrogeological criticality assessment, was then assigned to each threshold, with the aim of simplifying index signal interpretation by Civil Protection end-users. The forecast skill of both indices has been investigated at station level, through dichotomous and continuous statistical analysis, by comparing indices time evolution and hydrometric

level time series, taking advantage of typical assessment methods used in the signal theory, such as the derivative dynamic time warping. Moreover, spatial information given by both indices was assessed by comparing daily BDD and CAI stress maps and localization of effects at ground, collected from event reports, press releases and warnings. Obtained results indicated as the hydrological stress spatial information, highlighted by higher indices values, is coherent with the localization of affected municipalities and flood reports, while no stress overestimation is reproduced over those areas not involved in the event.

Objective dichotomous statistical analysis was performed over 78 hydrometric stations by using contingency tables, built by comparing indices and hydrometric height moderate threshold exceedances. Results evidenced high accuracy, with values exceeding 0.8 for both indices and all CSs. False alarm rates were under 0.5, while appreciable difference is given by the probability of detection ranging from 0.51 for CAI to 0.80 for BDD among the different case studies. Signal analysis has been carried out over 120 hours time series of indices and observed river stages. The DDTW over all stations was found to be

abundantly lower than 0.1, which is commonly referred as the threshold value beyond which to signals can be considered independent. Even if the stress signal behaviour is correctly reproduced by both indices, peak timing analysis shew some anticipation in signal peak occurrences, in the order of few hours. Timing bias is more pronounced for the CAI index, where displacements of more than -4 and up to about -7 hours are highlighted by all statistical parameters. As mentioned, validation scores were calculated considering all available hydrometers in the domain, however, it should be highlighted as many stations

among them, are placed downstream to dams. Therefore, in these points, flood propagation is heavily influenced by retention and release from artificial water storage, which is widespread over the considered geographical domain, heavily exploited in terms of hydroelectric power production. Indices performance would benefit of potential availability of retention and release data, however, the main aim of developing general thresholds for the proposed indices deals with contemplating data scarcity and hypothesis of unavailability of information about water uptake, which is very difficult to find, in the author's experience.

As for indices applicability, results highlighted a different indices response to different catchments and diverse flooding dynamics. According to Chen et al. (2010), floods may have different drivers: fluvial floods are mainly determined by the limited capacity of drainage systems, while pluvial floods are caused by deluges from river channels. However, the discrimination between a pluvial and a fluvial flood is not sharp; in the matter of fact, most events result in a combination of both processes. This condition often affects small hydrological basins, such as most Italian rivers (drained area lower than

10000 km$^2$, according to the definition provided by Chapman, 1992). According to the underlined differences found in CAI and BDD mapping, the proposed indices gives complementary information about hydrological stress over wide areas, as the former index appears to be more responsive to predominant pluvial flood dynamics affecting smallest tributaries, while higher stress identified by BDD occurs over main channels.



**Acknowledgements**

The present paper is written in the framework of the Agreement between the Centre of Excellence CETEMPS and the Functional Centre of Abruzzo Region. The authors would like to thank colleagues from the Civil Protection Dept. of Abruzzo for their feedback during the activities foreseen in the agreement. Moreover, part of this work is financed by the PON-AIM project, funded by the Italian Ministry of University and Research (MUR).


**Authors' contribution**

Conception and design: A. L., V. C., B. T.; Analysis: A. L., B. T., V. C.; Manuscript writing: A. L., V. C., B. T., M. V.; Draft revision: V. C., B. T., A. L.; Coordination: B. T.




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

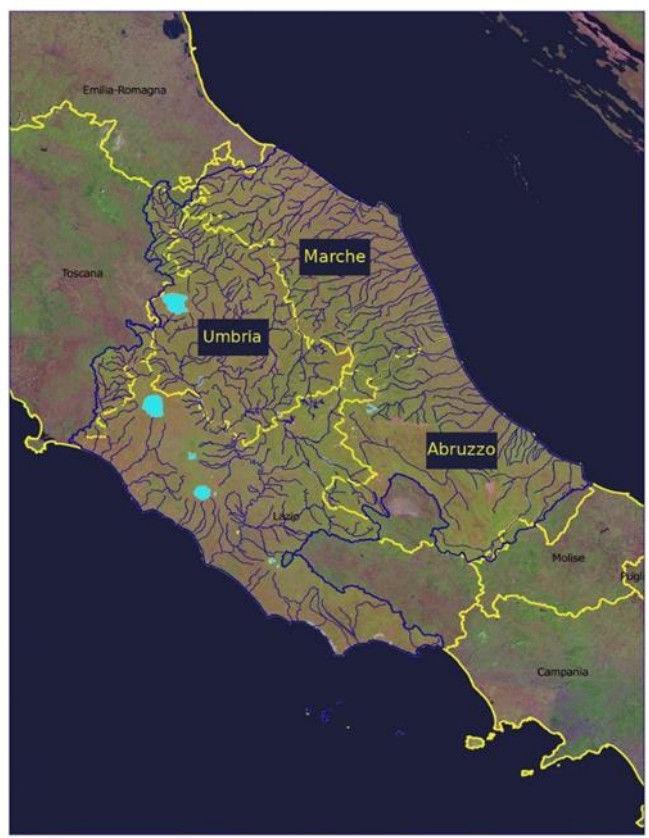

**Figure 1**: the Central Apennines Hydrological District (blue solid lines) and its main hydrography (blue thin lines). The north-eastern boundary is delimited by the Potenza river basin, while the south-eastern limit is represented by the Sangro basin in Abruzzo. The western side is delimited by the Tiber basin. Yellow lines indicates administrative boundaries of Italian regions (courtesy of Tiber Basin Authority, http://www.autoritadistrettoac.it/)




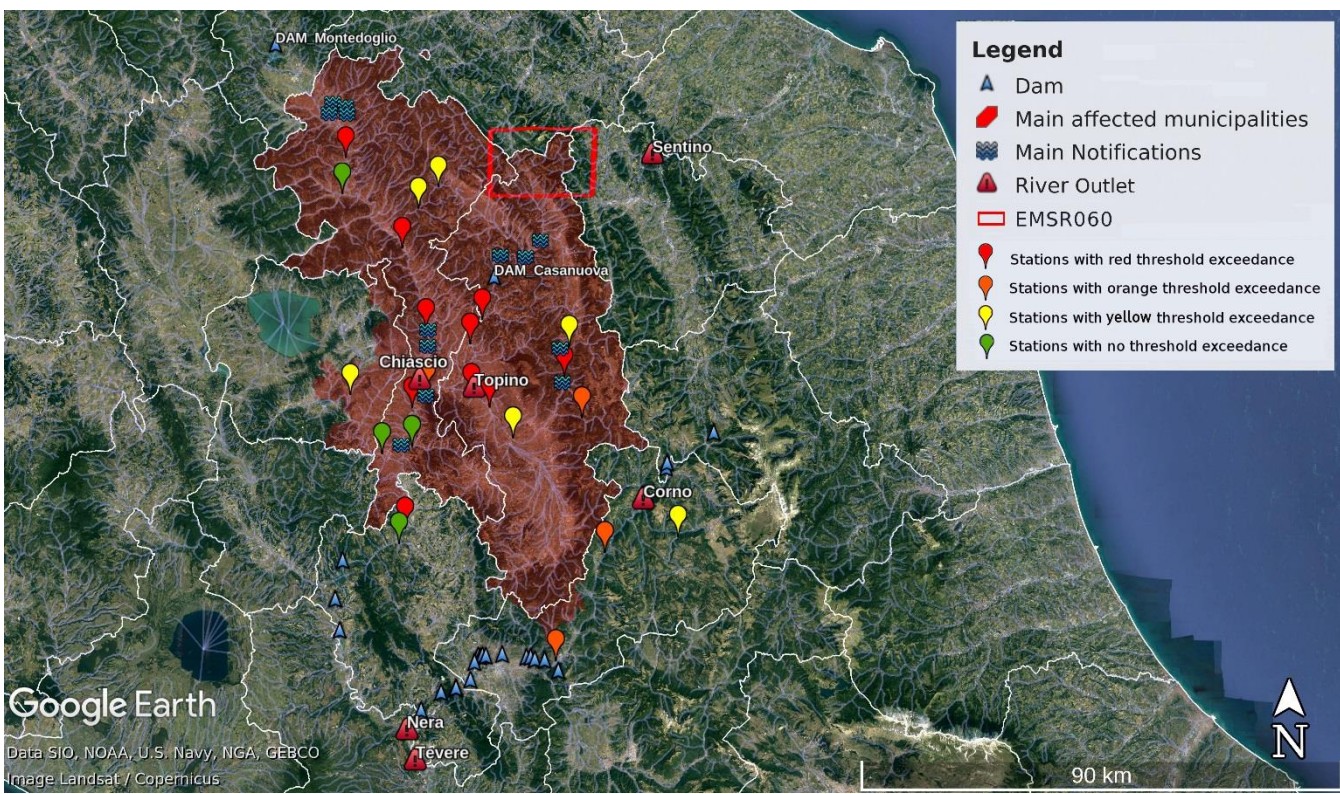

**Figure 2**: geo-referred information of ODB for CS01, Umbria Region, with localization of main recorded floods (blue waves), hydrometric
stations used for the indices validation (pinpoints). Red triangles indicate the position of outlets of main involved rivers, while blue triangles
indicates the presence of dams. Hydrometric station pinpoints are coloured according to the maximum hydrometric threshold reached during
the event. Municipalities areas affected by floodings are filled in red. Red rectangle represents the involved area published on COPERNICUS
Emergency Management Service Platform (https://emergency.copernicus.eu/mapping/list-of-components/EMSR060).






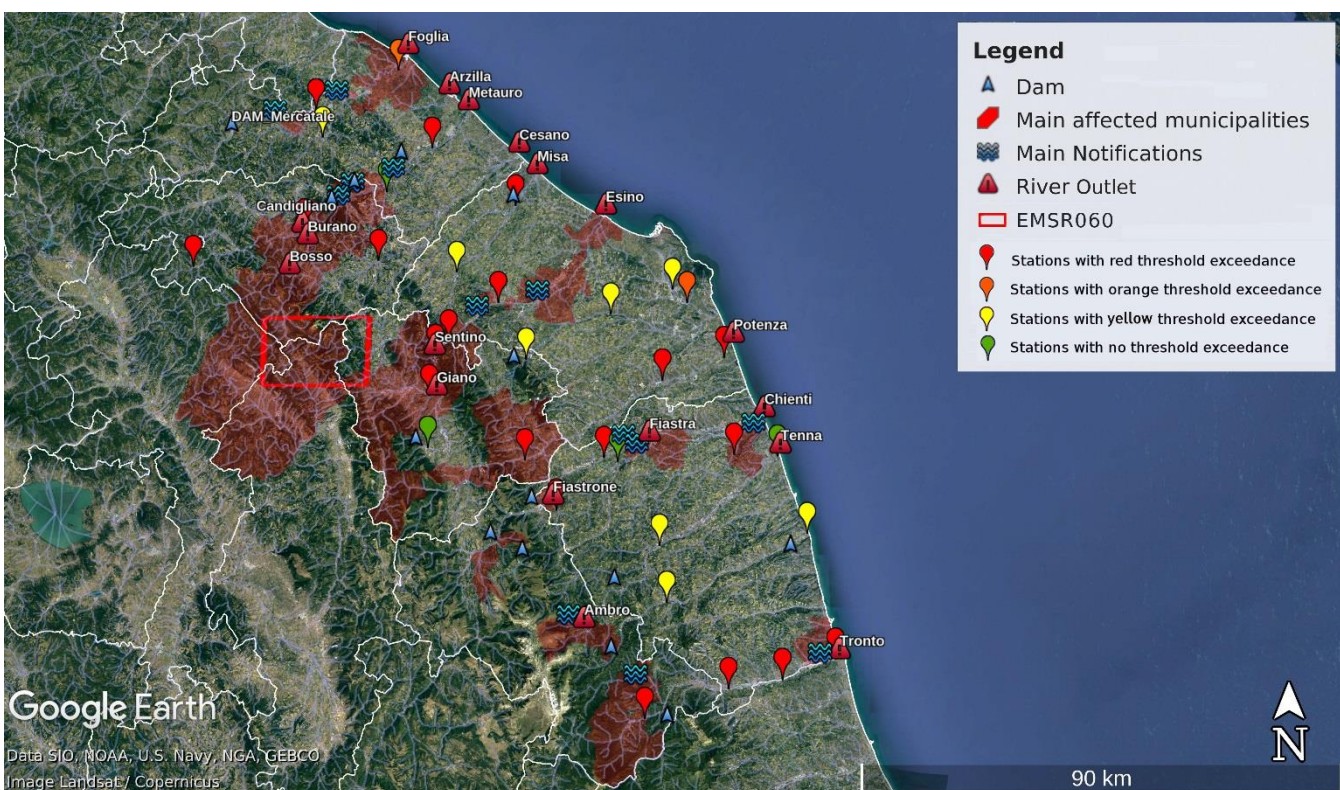

**Figure 3:** geo-referred information of ODB for CS02, Marche Region, with localization of main recorded floods (blue waves), hydrometric
stations used for the indices validation (pinpoints). Red triangles indicate the position of outlets of main involved rivers, while blue triangles
indicates the precence of dams. Hydrometric station pinpoints are coloured according to the maximum hydrometric threshold reached during
the event. Municipalities areas affected by floodings are filled in red.


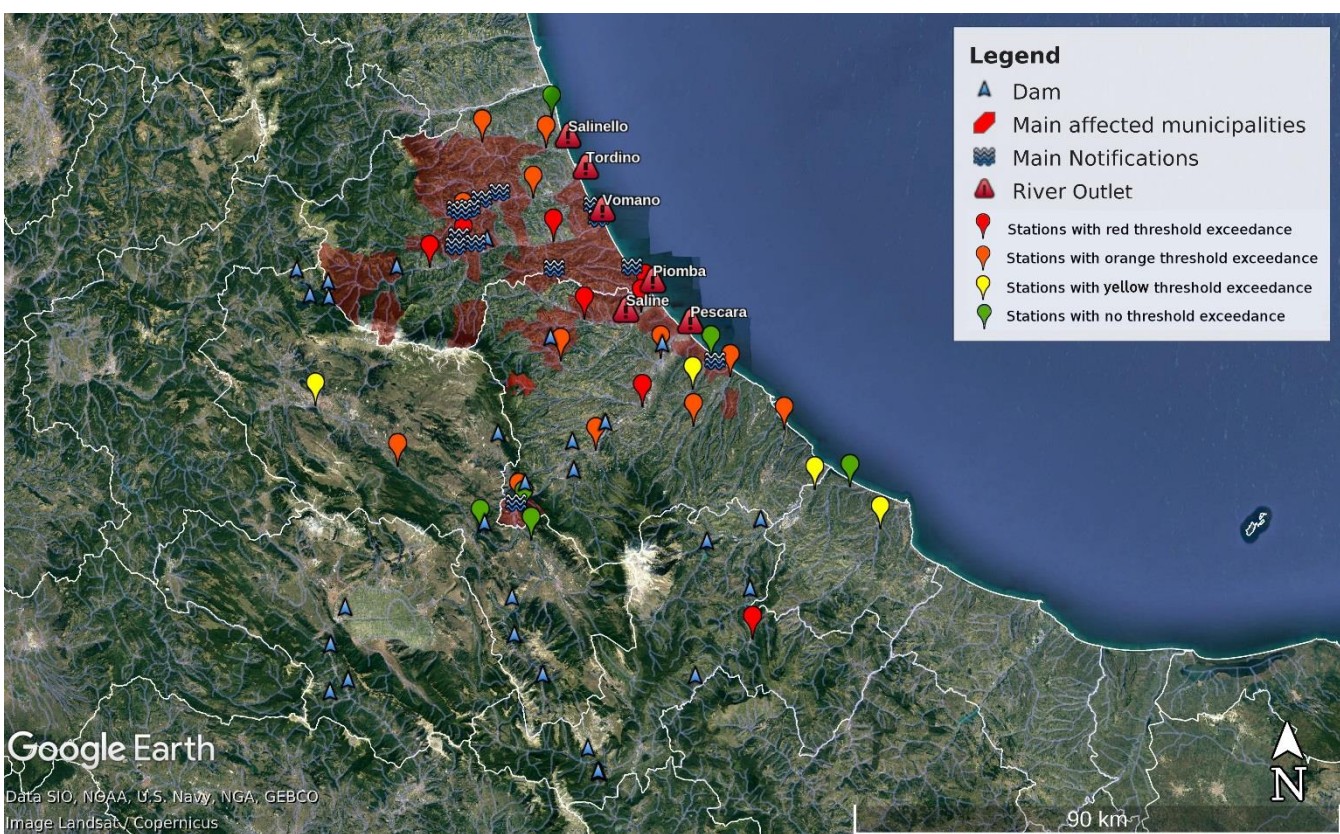

**Figure 4**: geo-referred information of ODB for CS03, Abruzzo Region, with localization of main recorded floods (blue waves), hydrometric
stations used for the indices validation (pinpoints). Red triangles indicate the position of outlets of main involved rivers, while blue triangles
indicates the presence of dams. Hydrometric station pinpoints are coloured according to the maximum hydrometric threshold reached during
the event. Municipalities areas affected by floodings are filled in red.




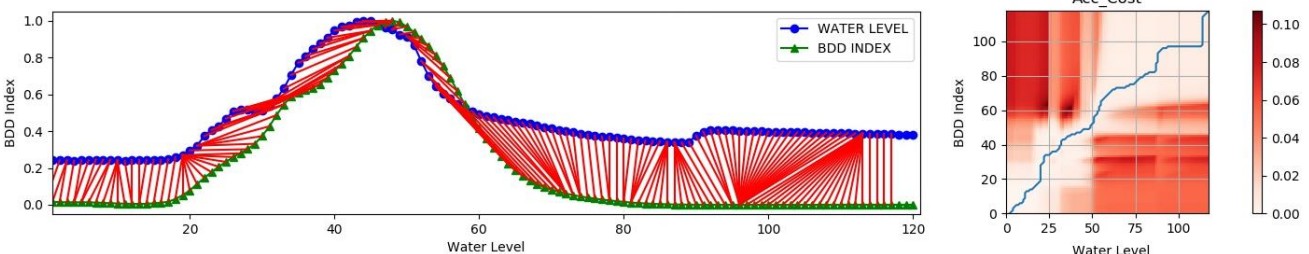

**Figure 5:** graphical representation of DDTW correspondences between two first derivatives of time series x(t) and y(t). In this case, time series are represented by two generic profiles of the hydrometric water level and the BDD index, at the same station point (from Keogh and Pazzani, 2001).


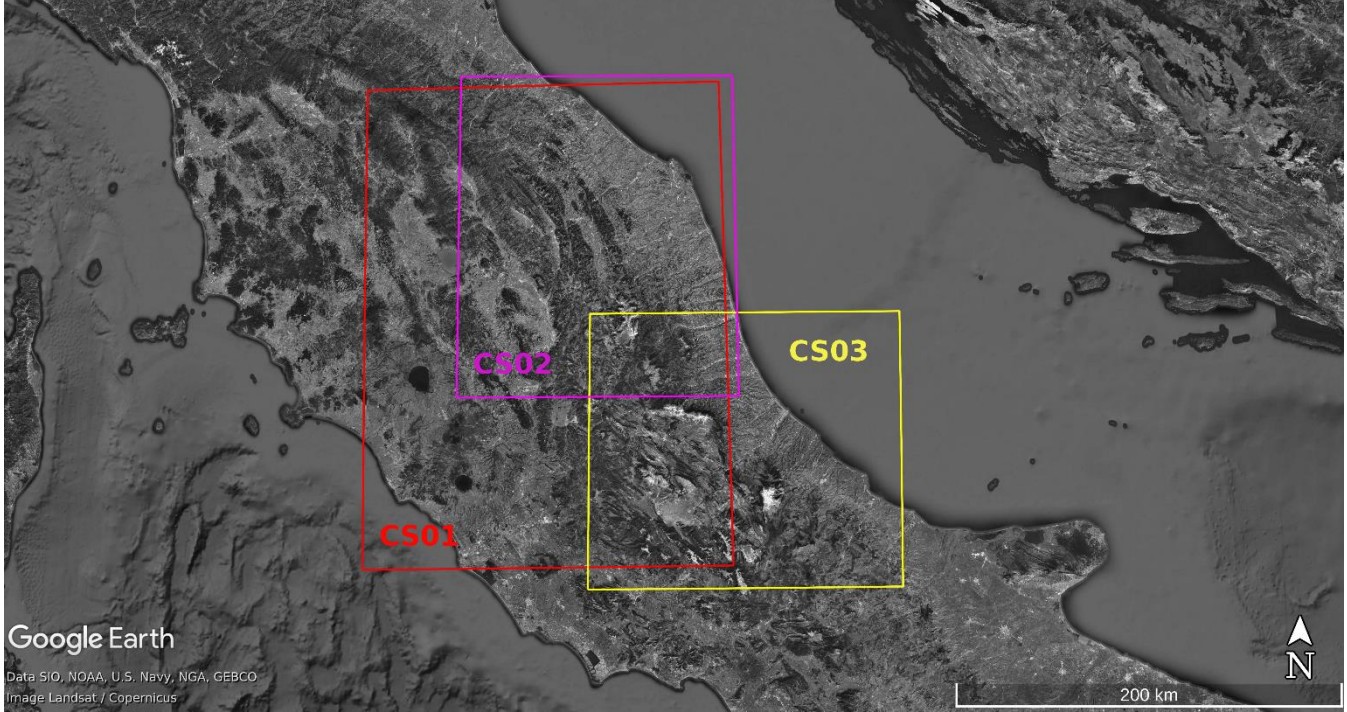

**Figure 6**: three CHyM geographical domains used for the simulation of the corresponding CSs. The red square encloses Umbria Region and the rest of Tiber basin for CS01, pink square refers to CS02 (Marche Region) and yellow square encompasses Abruzzo Region for CS03.






**Figure 7**: total accumulated rainfall (spatialization from rain gauges official network) during the event, from 11[st] November 2012 00 UTC to 13[rd] November 2013, 23 UTC (picture generated from the Dewetra Platform, Italian Civil Protection Department and CIMA Research
Foundation, 2014).


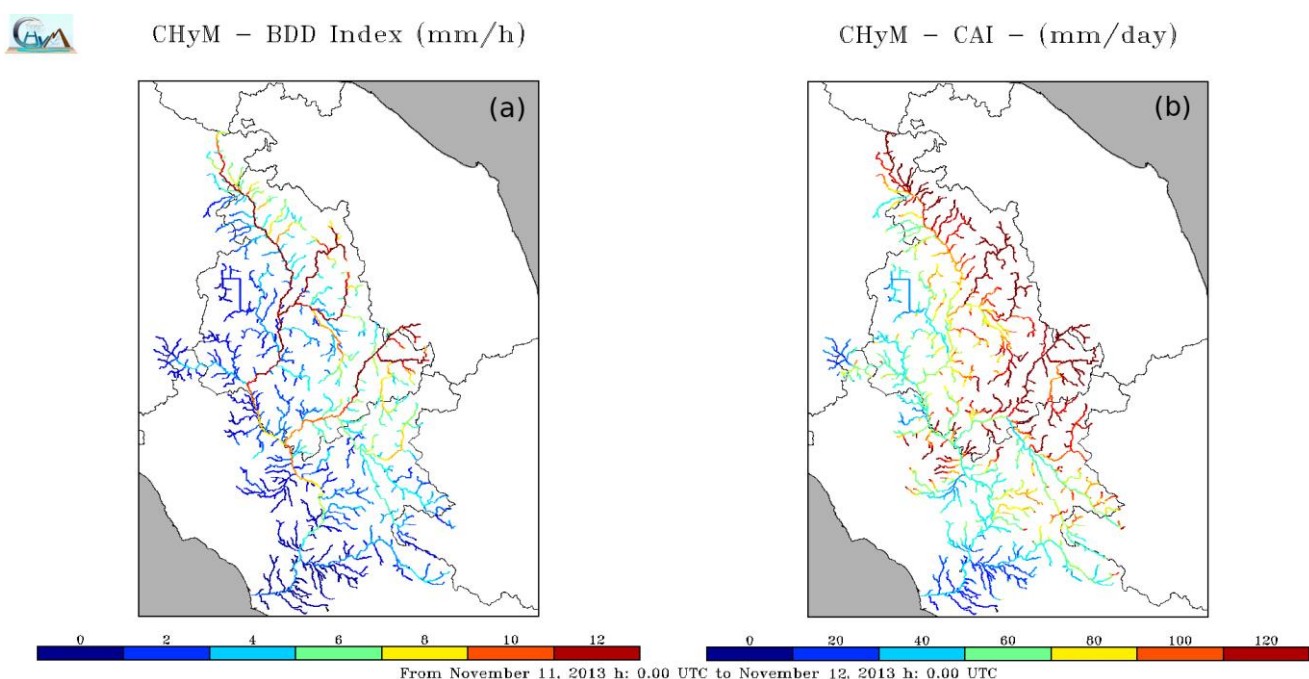

**Figure 8**: CS01 24 hours map of BDD index (left) and CAI index (right) obtained for November 11th 2013, by forcing the CHyM model with observed rainfall data. Warmer colours indicate river segments with higher flood stress. In both figure, the Umbria region drainage network, as well as the whole Tiber river basin, are highlighted.


**Figure 9**: time series of BDD and CAI indices, compared to hydrometric level observations for 6 relevant hydrometric stations chosen for CS01. Quantities profiles and related thresholds (flat lines) are normalized.


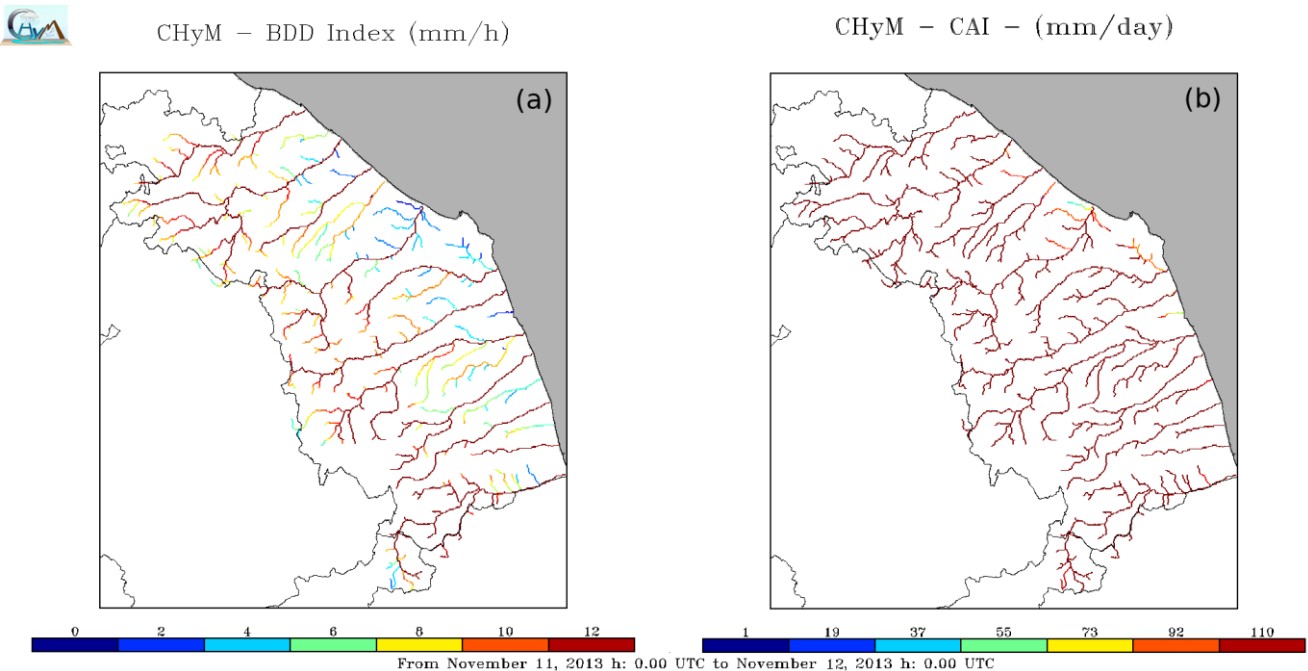

**Figure 10**: CS02 24 hours maps of BDD index (left) and CAI index (right) obtained for November 11th 2013, by forcing the CHyM model with observed rainfall data. Warmer colours indicate river segments with higher flood stress. In both figure, the Marche region drainage network is highlighted




**Figure 11**: time series of BDD and CAI indices, compared to hydrometric level observations for 6 relevant hydrometric stations chosen for CS02. Quantities profiles and related thresholds (flat lines) are normalized






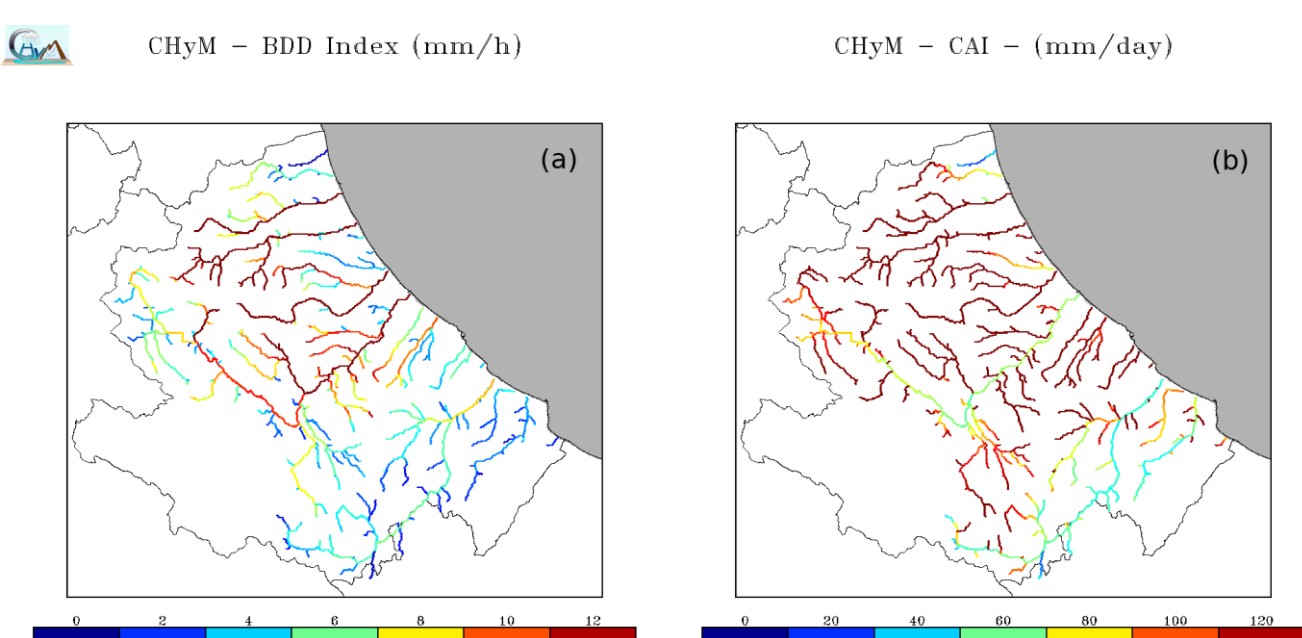

From November 12, 2013 h: 0.00 UTC to November 13, 2013 h: 0.00 UTC

**Figure 12**: CS03 24 hours map of BDD index (left) and CAI index (right) obtained for November 12th 2013, by forcing the CHyM model with observed rainfall data. Warmer colours indicate river segments with higher flood stress. In both figure, the Abruzzo region drainage network, belonging to the Central Apennine District, is highlighted



**Figure 13**: time series of BDD and CAI indices, compared to hydrometric level observations for 6 relevant hydrometric stations chosen for CS02. Quantities profiles and related thresholds (flat lines) are normalized.





**Table 1:** hydrogeological criticality levels officially defined by the Civil Protection Authorities. Regional Functional Centres define
hydrometric thresholds, in relevant river sections. Those thresholds are based on the return period concept, in order to individuate the
ciricality level to be assigned to the whole warning area (definitions conformed to Deliberation of Abruzzo Region Council no.659/2017,
Deliberation of Marche Region Council no. 148/2018 and Deliberation of Umbria Region Council no. 2312/2007 ).

| Threshold Colour-code | Hydrometric level | Criticality level | Description |
|---|---|---|---|
| Green | Below THR1 | Absence of significant predictable phenomena | Regular Criticality Level, possible local floods due to non-sufficient drainage of meteoric waters |
| Yellow | Above THR1 (Yellow threshold) | Ordinary Criticality | Ordinary Criticality Level: Weak flow peak. Water level values corresponds to low water level and generally below the natural terrain level. |
| Orange | Above THR2 (Orange threshold) | Moderate Criticality | Moderate Criticality Level. Flow peak with limited erosion and transport. Water levels corresponds to the floodplain and river expansion to the levee. The natural floodplain is exceeded. |
| Red | Above THR3 (Red Threshold) | High criticality | High Criticality Level. Significant discharge peak and diffused erosion and transport. Water Level corresponds to the whole riverbed. |






**Table 2:** summary of relevant damages reported for each Case Studies and used information sources.

| Case Study | Date | Region | Reported damages | Information sources | | |
|---|---|---|---|---|---|---|
| | | | | OR | PR | V |
| **CS01** | 11-12 Nov 2013 | Umbria | Interruption of several roads and bridges, isolated villages, damage to buildings and roads, a hospital isolated. | ✓ | ✓ | ✓ |
| | | | ***Important notes from OR*** | | | |
| | | | The large dams in the Tiber basin (Montedoglio and Corbara on Tiber and Casanuova on Chiascio river) played a crucial role for the storage of upstream incoming volumes, allowing the lamination and the misalignment of the full floods downstream. | | | |
| **CS02** | 11-12 Nov 2013 | Marche | Interruption of several roads, houses evacuated, isolated villages and two fatalities. | ✓ | ✓ | ✓ |
| | | | ***Important notes from OR*** | | | |
| | | | The large dams in the Foglia, Metauro, Chienti and Tronto basins played a crucial role for the storage of upstream incoming volumes and allowed the lamination and the misalignment of the full floods downstream. | | | |
| **CS03** | 12-13 Nov 2013 | Abruzzo | Flooding phenomena affected the small Abruzzo Rivers. Interruption of several roads, damage to buildings and roads. | X | ✓ | ✓ |
| **Legend: OR:** Official Civil Protection Report; **PR:** Press releases; **V:** Videos | | | | | | |







**Table 3**: contingency table structure used for the validation analysis.

|  |  | Observed | |
| --- | --- | --- | --- |
|  |  | Yes | No |
| Estimated | Yes | Hit (H) | False Alarm (FA) |
|  | No | Miss (M) | Correct Negative (CN) |


**Table 4:** CHyM domain set-up for the analysed case studies. Please, note that the rain gauges data may not be all available during the entire event, due to interruption of electric supply.

|  | CS01 | CS02 | CS03 |
| --- | --- | --- | --- |
| **Horizontal Resolution** | 370 m | 270 m | 270 m |
| **Domain dimension (nlon*nlat)** | 750x550 | 650x550 | 710x470 |
| **No. of hourly timesteps** | 240 | 240 | 240 |
| **No. of rain gauges in the domain** | Up to 371 | Up to 138 | Up to 135 |
| **No. of hydrometric stations used in the domain** | 22 | 28 | 26 |






**Table 5**: CAI and BDD indices scores for all CSs. Values for single CS are averages calculated over all hydrometric stations
located in the domain.

| | CR | A | POD | FAR | LTP | RLTP | CTD | DDTW |
|---|---|---|---|---|---|---|---|---|
| **BDD** | | | | | | | | |
| CS01 | 0.86 | 0.88 | 0.70 | 0.46 | -0.6 | -0.05 | -1.0 | 0.02 |
| CS02 | 0.75 | 0.85 | 0.68 | 0.37 | 0.4 | 0.04 | -1.4 | 0.04 |
| CS03 | 0.77 | 0.91 | 0.80 | 0.48 | -8.6 | -1.11 | -4.2 | 0.09 |
| **TOT** | **0.79** | **0.88** | **0.72** | **0.43** | **-2.9** | **-0.37** | **-2.2** | **0.05** |
| **CAI** | | | | | | | | |
| CS01 | 0.77 | 0.88 | 0.51 | 0.44 | -8.5 | -0.67 | -7.4 | 0.06 |
| CS02 | 0.75 | 0.82 | 0.55 | 0.43 | -4.8 | -0.56 | -5.6 | 0.05 |
| CS03 | 0.77 | 0.93 | 0.72 | 0.40 | -12.6 | -1.59 | -7.62 | 0.11 |
| **TOT** | **0.76** | **0.88** | **0.59** | **0.42** | **-8.6** | **-0.94** | **-6.9** | **0.07** |