# Peer review of "User-oriented hydrological indices for early warning system. Validation using post-event surveys: flood case studies on the Central Apennines District."

_Hydrology and Earth System Sciences, 2020_

## Referee Comment (RC1) · Anonymous Referee #1 · 27 Aug 2020

General comments

This paper presents an application of two new indexes for flood prediction in central Italy. Indexes are linked to two different flooding sources: a pluvial index and a fluvial flood index. The case study is the November 2013 event that hit Central Appennines in Italy. I found the topic of paper very interesting to HESS readers but some adjustments are needed before publication. The main concern is about hydrological model calibration. Authors mention necessity of calibrating some parameters but it is not clear if the CHym model has been calibrated before its application.

[Figure]

Specific comments

L50-52: "In the EU Directive 2007/60/CE concerning the "Assessment and management of flood risks", the realization of a flood risk map is foreseen over river basins with a significant potential risk of flooding (European Parliament, 2007). Prediction of flood events is therefore important to enhance mitigation strategies to face hydrological events." It is not clear the connection between flood risk map and prediction of flood events. They are two distinct concepts not connected necessarily. Flood risk maps are assessed offline based on scenario events (for given return periods). Flood prediction is used in real time to forecast in advance the arrival of flood. This is not the only possible measure. There are structural measures to consider as well.

L106 Spatial resolution of hydrological model is 90m. Table 4 presents further spatial resolution different according to case study. Please clarify. How model parameters are scaled, if any, when spatial resolution changes?

L146-148. This sentence states that Mn value should be calibrated but the final resulting value is not explained.

L217-220 Is an initial value of water in the two reservoirs considered? This is linked to model spin-up. Are there any parameters to calibrate for infiltration computation?

L 243 Authors stress on the necessity for long time series of flow discharge data and present the proposed approach as a means to overcome that problem. But the presented system is based on the CHyM hydrological model that, in turn, needs calibration, I suppose. Please clarify the real advantages of the proposed approach as regards model calibration.

L242-266. Consider moving this part to Introduction section.

L285-290 It is not clear how corrivation time is computed for CAI computation. It seems the sum of time to pass through different river reaches. It sit the time for passing the longest flow path? How velocity of single river reach is computed? It should change

with roughness and slope.

L295-300 Authors present three rainfall intensity as warning thresholds and say they are defined with empirical tests. The readers interested in applying this procedure to other sites should know how to define rainfall threshold value. Are they universal for all basins in the world? The same consideration applies for BDD index. At line 404 authors state that "the calibration of the indices thresholds was chosen in order to maximize the hit rate..", please clarify.

L311 why ordinary index present two values?

L527 Is a spin-up time of 120 hours enough for model initialization? See comment about infiltration model.

L565 Can overestimation be explained by a lack of flood damage information in that area?

---

## Referee Comment (RC2) · Anonymous Referee #2 · 11 Sep 2020

The paper presents an interesting method that could be used in the practice to mitigate flood risk. Despite the interesting topic, the paper needs to address some issues before being ready for publication. I'll mention those not already covered by the other reviewer.

In the abstract the effect of dams is mentioned, however, there is not either a further discussion about it or presentation of results regarding dams. The paper would benefit from a reorganization. For instance, the study area description cannot be part of results but should be a separated sect. or subsection. Sect. 4 comprehends a part that could be considered belonging more to an Introduction than to a Method section. I would also

kindly suggest to add a discussion section to provide a brief discussion on the results that otherwise are just presented in the Results section and then summarized in the Conclusions section. I kindly suggest to move the stress indices description under the section Materials and Methods. Please, recall the figures in consecutive order. In case, move some figures before the others. The English grammar should be revised. Even though the topic is relevant and of interest also considering as end-user the Civil Protection, I wonder if the topic would be more suitable on NHESS for instance or on a more technical Journal. In the following, there are some more issues to be addressed.

line 85: since the indices "are calibrated taking into account a correspondence between the issued civil protection alarm level and index threshold." I ask to clarify how the lead time is taken into account. The lead time available for triggering civil protection measures is typically short, especially in case of flash floods. The longer this time is, the better.

line 149 p. 5: please notice that it is km and not Km.

Table 2: after "each", the singular form is needed.

line 463: replace "figura" with "figure".

line 491 "throught" instead of "though".

line 285: it is not statistically relevant that often floods and flash floods occur in un-gauged basins. This sentence may refer to the need of improving the sensors networks deployment or to the fact that ungauged basins would largely benefit from the method proposed.

line 345 the time to peak should be considered proportional to the concentration time as 3 h may be a long time for small basins.

p.10: for the sake of understanding, I kindly suggest to remove Table 3 and add the explanation of acronyms before introducing the indices.

line 469: Casanuova River? Maybe Chiascio River and Casanuova dam.

line 620: "two signals" instead of "to signals"

Figures Figures 1 and 7 are in low quality Captions should be self-standing, please explain all acronyms present both in the figure and in the legend. Figs. 9, 11 and 13 are not clear as the legends are not explained in the caption.

---

## Author Comment (AC1) · 23 Oct 2020

1. "General comments. This paper presents an application of two new indexes for flood prediction in central Italy. Indexes are linked to two different flooding sources: a pluvial index and a fluvial flood index. The case study is the November 2013 event that hit Central Appennines in Italy. I found the topic of paper very interesting to HESS readers but some adjustments are needed before publication. The main concern is about hydrological model calibration. Authors mention necessity of calibrating some parameters but it is not clear if the CHym model has been calibrated before its application."

Response: The hydrological model has been widely calibrated using climatological discharge time series of the Po river, as reported in Coppola et al. (2014). To this aim, it is important to note that the conditions of the Po River are representative of many alluvial rivers in Europe (Di Baldassarre et al., 2009). As stressed out in the paper, the absence of updated discharge estimates in many Italian regions makes difficult to calibrate the model specifically for each basin. Starting from the climatological calibration on the Po basin and considering the civil protection operational purposes, aimed at identifying river flow conditions where significant discharges are observed, stress indices are introduced also to overcome the general calibration issues. The calibration focus was then moved from the classical scope of the "best prediction of the discharge amount", toward a new approach, where the hydrological stress as a whole is predicted and validated. In this context, discharge is part of the hydrological stress, but is related to other parameters, such as hydraulic radius (in BDD), and catchment concentration time (in CAI), that plays also an important role in flood dynamics.

2. "L50-52: "In the EU Directive 2007/60/CE concerning the "Assessment and management of flood risks", the realization of a flood risk map is foreseen over river basins with a significant potential risk of flooding (European Parliament, 2007). Prediction of flood events is therefore important to enhance mitigation strategies to face hydrological events." It is not clear the connection between flood risk map and prediction of flood events. They are two distinct concepts not connected necessarily. Flood risk maps are assessed offline based on scenario events (for given return periods). Flood prediction is used in real time to forecast in advance the arrival of flood. This is not the only possible measure. There are structural measures to consider as well."

Response: we agree with this comment. The information provided by the CHyM operational stress indices maps is complementary with flood risk maps, therefore, both information needs to be taken into account by the civil protection operator for his/her evaluation, when a flood event is expected to occur. In our opinion, simulation and prediction of flood events is connected with the implementation of mitigation strategies,
because it can be a useful pre-requisite for the mitigation planning phase. Considering that the paper is not clear in this point, we propose to replace the sentence with the following: "... In the EU Directive 2007/60/CE concerning the "Assessment and management of flood risks", the realization of a flood risk map is foreseen over river basins with a significant potential risk of flooding (European Parliament, 2007). To this aim, tools for flood events prediction may also provide useful information for the mitigation strategies planning phase."

3. "L106 Spatial resolution of hydrological model is 90m. Table 4 presents further spatial resolution different according to case study. Please clarify. How model parameters are scaled, if any, when spatial resolution changes?"

Response: resolution of 90 m is actually the resolution of the NASA SRTM DEM source file (https://lpdaac.usgs.gov/products/srtmgl3v003/) which is implemented in the model. For this reason, the CHyM model can perform simulation with horizontal resolutions ≥ 90 m. For our national operational activity, we had divided the Italian territory in 7 geographical sub-domains, each domain has its own spatial resolution, chosen in order to optimize computational requirements (lower resolutions means faster simulations) and the correct drainage network rebuilt (higher resolutions means more accurate drainage network reconstruction). In this paper, we maintained the operational spatial resolution associated to each sub-domain. Starting from the NASA data, the DEM is upscaled by applying the Cellular Automata spatial interpolation technique. All those information are contained in Coppola et al. (2014).

4. "L146-148. This sentence states that Mn value should be calibrated but the final resulting value is not explained."

Response: the Mn value can be calibrated if needed, since it was reported in the model as a parameter. For our simulations, we are using the default value of 4.5.

5. "L217-220 Is an initial value of water in the two reservoirs considered? This is linked to model spin-up. Are there any parameters to calibrate for infiltration computation? "

Response: in our simulations, the initial value of water in the two reservoirs is not considered, because no data are provided about release and withdrawals of water from the water reservoirs. The spin-up of the model is set to 5 days to reproduce initial flow conditions. Due to the lack of water storage data, it is not possible to properly assess the flow discharge simulation, therefore, we can only state that the discharge simulation from our model differs from observations and highlight the presence of an anthropic impact due to the presence of water reservoirs upstream. According to our experience, we have found that indices peak timing and their shifts respect to observed hydrometric level can provide information about the flood management through water reservoirs release and withdrawals, that are able to postpone (or anticipate) discharge maxima propagation downstream. The infiltration computation is explained in Coppola et al 2014 at paragraphs 3.4 and 3.5. The same parameterizations are used in this work.

6. "L 243 Authors stress on the necessity for long time series of flow discharge data and present the proposed approach as a means to overcome that problem. But the presented system is based on the CHyM hydrological model that, in turn, needs calibration, I suppose. Please clarify the real advantages of the proposed approach as regards model calibration. "

Response: we thank the reviewer for this comment, that gives us the possibility to better stress on our findings. We have partially replied to this observation in our response to the general comment. In general, long time series of flow discharge data are necessary to calibrate and validate hydrological models. However, such data are not always available from all Italian regions and, in many cases, rating curves used for the discharge estimation starting from the hydrometric level are not constantly updated. As stated by the WMO, hydrometric level is also a strongly non-stationary parameter. Furthermore, hydrometric level measurements are not available for major floods, when sensors installed along rivers stops to work due to severe meteorological conditions. For this reason, many data in the upper part of the rating curve are missed and larger

errors in discharge estimation are often associated to higher discharge bins. Finally, hydrometers are installed over main river channels and small catchments are often excluded from discharge estimations, even if they are more prone to destructive flooding phenomena, especially in a complex orography context. Hydrometric/discharge thresholds are defined punctually and differs on each sensor. In our stress indices approach, discharge and runoff are combined with geographical information related to the upstream basin displacement, through the use of other variables, such as the hydraulic radius (a function of the drained area) and concentration time (that implicitly consider runoff conditions upstream), therefore, they are able to give information in each point of the drainage network and their mutual variation from upstream to downstream along the river path is proportional. For this reason, general thresholds, valid in all grid-points of the drainage network may be defined. Moving from discharge to combined discharge-based and runoff-based indices, with the aim of calibrating such indices on threshold-basis for flood alert purposes, gives us the possibility to calibrate and validate a different information, which is not the discharge amount, but the river stress conditions, which is qualitatively given by civil protection authorities through the use of hydrometric thresholds, as well as stress timing. Furthermore, the good estimate of the stress state on a river channel is also provided by event reports and from press releases in those locations where no sensors and, hence, no threshold are defined. Since the indices validation is not numerical, the problem of missing discharge data is overcome, being the threshold-based calibration a sufficient condition for our purpose to validate an alert system, rather than physical quantities.

7. "L242-266. Consider moving this part to Introduction section."

Response: DONE, insert in line 55

8. "L285-290 It is not clear how corrivation time is computed for CAI computation. It seems the sum of time to pass through different river reaches. It sit the time for passing the longest flow path? How velocity of single river reach is computed? It should change with roughness and slope."

Response: the time of concentration is computed for each grid-point of the geographical domain. It can be defined as the time required to a raindrop to travel from the hydraulically most distant point in the watershed to the outlet. The outlet must be intended in the numerical sense; namely, it may be a "mouth cell" draining toward a sea point, a "tributary mouth cell" draining toward the interception with the main river or a cell draining toward border of the simulated domain. The water velocity for each cell of the domain is computed according the equation [2.1.3] written in paragraph 2.1. The velocity computation considers the acclivity, estimated as the sinus of the terrain slope in the direction of surface flow, as well as the roughness, through the use of the Manning's roughness coefficient depending on the land use cover. For example, the largest catchment in Abruzzo region, the Aterno-Pescara is simulated to have an upstream area of 3310 km2 and a concentration time of 20 hours, approximately. The concentration time used in the CAI calculation is an average calculated on all possible concentration times resulting from draining paths toward the considered grid-point.

9. "L295-300 Authors present three rainfall intensity as warning thresholds and say they are defined with empirical tests. The readers interested in applying this procedure to other sites should know how to define rainfall threshold value. Are they universal for all basins in the world? The same consideration applies for BDD index. At line 404 authors state that "the calibration of the indices thresholds was chosen in order to maximize the hit rate..", please clarify. "

Response: although the units of measurement of the indices are expressed in mm, they do not represent rainfall. Actually both indices refer to the water accumulated on the ground over the time. Three different thresholds for each of the two indices have been defined, in accordance with the protocols in use at the national civil protection department. Since our intention is to develop unique thresholds, having the same values in all grid-points, we had to optimize threshold choice in order to maximize hit rate and minimize false alarms. For the definition of indices thresholds, we decided to assign values maximizing the hit rate scores, i.e., we have chosen indices values

causing a slight increase of the false alarm rate to also maximize the hit rate. In order to avoid some further burdening of this paper, only results related to the moderate threshold (orange, pre-alert) threshold are reported. The reason of our preference of this particular threshold lays on the consideration of its meaning in the civil protection alert system. In fact, the orange threshold exceedance can be considered the most crucial one for the civil organization, because its exceedance starts the activation of protection measures for people and infrastructure safety, as foreseen in risk plans. As for the "universality" of our indices, our main purpose is to avoid developing different thresholds on different areas, for this reason, we had tested them over a wide area in Central Italy, where many different catchments are located. It is untimely to say that indices are universally applicable, but we are confident to be able to extend our validation to other areas in Italy and Europe, due to results we are having in our ongoing research.

10. "L311 why ordinary index present two values? "

Response: Thanks for your observation, it is a typo.

11. "L527 Is a spin-up time of 120 hours enough for model initialization? See comment about infiltration model."

Response: given the small extension of the involved catchments, 120 hours of initializations seems to be enough for the model initialization. Moreover, it should be noticed that stress indices are used to detect hydrological situations where relevant discharges, driven by significant rainfall events in short time (few hours to few days) are present.

12. "L565 Can overestimation be explained by a lack of flood damage information in that area?"

Response: we really thank the reviewer for this comment. We totally agree with this assertion: very likely, the overestimation also depends on a lack of information. To be honest, in these circumstances and without evidences, we can only say that the model

did not properly simulate hydrological stress.

---

## Author Comment (AC2) · 23 Oct 2020

The paper presents an interesting method that could be used in the practice to mitigate flood risk. Despite the interesting topic, the paper needs to address some issues before being ready for publication. I'll mention those not already covered by the other reviewer.

1. "In the abstract the effect of dams is mentioned, however, there is not either a further discussion about it or presentation of results regarding dams."

Response: thank you for your careful observation. We have deleted the sentence

relating dams because it was redundant.

2. "The paper would benefit from a reorganization. For instance, the study area description cannot be part of results but should be a separated sect. or subsection. "

Response: As suggested, the study area description is a new section, now.

3. "Sect. 4 comprehends a part that could be considered belonging more to an Introduction than to a Method section."

Response: we agree with this consideration, however, we found that providing such information here makes this section self-consistent. Considering the complexity of the paper, in our opinion it can facilitate the understanding of the work done.

4. "I would also kindly suggest to add a discussion section to provide a brief discussion on the results that otherwise are just presented in the Results section and then summarized in the Conclusions section. "

Response: this is a right observation. Actually, during the drafting of the paper, we had conceived to insert a single section treating both the results and their discussions with the aim of not further lengthening the paper.

5. "I kindly suggest to move the stress indices description under the section Materials and Methods. Please, recall the figures in consecutive order. In case, move some figures before the others."

Response: as suggested by the Reviewer, we moved the indices description.

6. "The English grammar should be revised."

Response: The manuscript has been revised and some corrections have been made. Any further corrections from the reviewer are welcome.

7. "Even though the topic is relevant and of interest also considering as end-user the Civil Protection, I wonder if the topic would be more suitable on NHESS for instance

or on a more technical Journal. In the following, there are some more issues to be addressed."

Response: the reviewer is right when he/she stresses out that the developed indices are a tool definitely intended for civil protection end users. This is one aspect of our research; the other relevant aspect is the validation methodology we had proposed, that deals with lack of discharge data and homogeneous and well organized flood databases. Moreover, some validation techniques and are typical of other scientific fields and have been adapted and applied to hydrometeorology. For this reason we believe that this work is suitable for both journals.

8. "line 85: since the indices "are calibrated taking into account a correspondence between the issued civil protection alarm level and index threshold." I ask to clarify how the lead time is taken into account. The lead time available for triggering civil protection measures is typically short, especially in case of flash floods. The longer this time is, the better. "

Response: The calibration of the indices thresholds related was carried out by a comparison with hydrometric threshold and water level profile behavior. This work is aimed at validating indices information in "perfect conditions" i.e. with forcing the hydrological model with observed meteorological model variables. In our operational activity, as stressed out in Ferretti et al. (2020) and Colaiuda et al. (2020), the CHyM model uses meteorological observed data for the spin-up process and meteorological model output to predict hydrological stress for the next 24/48 hours. Therefore, operationally, when we release the hydrological forecast, we give this information from 6 to 48 hours in advance.

9. "line 149 p. 5: please notice that it is km and not Km. " Response: Done

10. "Table 2: after "each", the singular form is needed." Response: Done

11. "line 463: replace "figura" with "figure". " Response: Done

12. "line 491 "throught" instead of "though". " Response: Done

13. "line 285: it is not statistically relevant that often floods and flash floods occur in ungauged basins. This sentence may refer to the need of improving the sensors networks deployment or to the fact that ungauged basins would largely benefit from the method proposed. "

Response:

Unfortunately, especially in Italy, it is statistically significant that floods and flash floods occur in unmeasured basins because the monitoring network is poor. The considered sentence may refer to the fact that non-instrumented basins would greatly benefit from the proposed method. Other authors also suggest that the index approach may be a possible solution (e.g. Alfieri et al, 2012 highlighted as precipitation-based indices are preferable over uninstrumented rivers). In detail we have referred to the following assessment reported by Alfieri et al., 2017: "Flash floods usually occur in ungauged catchments, where the only source of information is post-event descriptive reports. Besides, even when gauging stations are available, they are sometimes damaged and made inoperative by the rage of the flood flow." The reference is also insert in the paper. There is a mistake concerning the year of publication which is 2017 instead of 2019.

14. "line 345 the time to peak should be considered proportional to the concentration time as 3 h may be a long time for small basins. "

Response:

correct observation. this information is provided in the calculation of the RLTP (Relative Leg Time Peak) score. We agree that three hours could not be suitable for the smallest catchment. In our collaboration with the Civil Protection Dept., "3 hours" of guard band are anyway the minimum time needed to activate protection measurements in case of emergency.

15. "p.10: for the sake of understanding, I kindly suggest to remove Table 3 and add the explanation of acronyms before introducing the indices."

Response:

we agree with the comment, however, during our experience in conferences we noticed as, although it seems intuitive to understand how the contingency table is structured, this is missing in some cases. Since it takes up little space in the paper, we preferred to report this information as a table, which seems to be easier to consult.

16. "line 469: Casanuova River? Maybe Chiascio River and Casanuova dam." Response: Done

17. "line 620: "two signals" instead of "to signals". " Response: Done

18. "Figures 1 and 7 are in low quality Captions should be self-standing, please explain all acronyms present both in the figure and in the legend. "

Response: Done

Caption Figure 1: the Central Apennines Hydrological District (blue solid lines) and its main hydrography (blue thin lines). The north-eastern boundary is delimited by the Potenza river basin, while the south-eastern limit is represented by the San-gro basin in Abruzzo. The western side is delimited by the Tiber basin. Yellow lines indicates administrative boundaries of Italian regions. The three considered regions were highlighted: Umbria, Marche, Abruzzo (courtesy of Tiber Basin Authority, http://www.autoritadistrettoac.it/).

Caption Figure 7: total accumulated rainfall (spatialization from rain gauges official network) during the event, from 11st November 2012 00 UTC to 13rd November 2013, 23 UTC (picture generated from the Dewetra Platform, Italian Civil Protection Department and CIMA Research Foundation, 2014). The localization of the six raingauges were indicated on the map: 1) Castel del Monte station (Abruzzo Region), 2) Castelluccio di Norcia station (Umbria Region), 3) Conca 1 station (Marche Region), 4) Gualdo Tadino

station (Umbria Region), 5) Pintura di Bolognano station (Marche Region); 6) Pretara station (Abruzzo Region). The raingauges recorded significant accumulated rain (up to 400 mm per 72 hours, purple area).

19. "Figs. 9, 11 and 13 are not clear as the legends are not explained in the caption."

Response: Done

Caption Figure 9, 11 and 13: time series comparison for six hydrometric stations: BDD hourly profile (red line), BDD Moderate Threshold (red flat line); CAI hourly profile (green line), CAI Moderate Threshold (green flat line); Hydrometric Level hourly profile (blue line), Hydrometric Level Moderate Threshold (blue flat line). Quantities profiles and related thresholds are normalized.

---

## Author Response (AR2)

**Author's Response**

**Anonymous Referee #2 Submitted on 15 Feb 2021**

Many thanks for the suggestions, corrections have been made.

1. Line 726 p. 24: "it is baed". **DONE**
2. Line 774 p. 25: the sentence seems truncated, could you please give a hint about its calculation (e.g. "it is function of…")? **DONE**
3. Line 907 p.30. if you are referring to the figures 1,2,3, then it's "maps" **DONE**
4. Please note that the figures should be recalled in ascending order, please rename figures where necessary. **DONE**